# Insights into substrate recognition and specificity for IgG by Endoglycosidase S2

**Asaminew H. Aytenfisu**[1], **Daniel Deredge**[1], **Erik H. Klontz**[2], **Jonathan Du**[2,3], **Eric J. Sundberg**[2,3], **Alexander D. MacKerell, Jr**[1] *

1 University of Maryland Computer-Aided Drug Design Center, Department of Pharmaceutical Sciences, School of Pharmacy, University of Maryland, Baltimore, Maryland, United States of America, 2 Department of Medicine, University of Maryland School of Medicine, Baltimore, Maryland, United States of America, 3 Department of Biochemistry, Emory University School of Medicine, Atlanta, Georgia, United States of America

* alex@outerbanks.umaryland.edu

**Data Availability Statement:** All programs used in support of this publication are publically available from http://openmm.org/, https://academiccharmm.org/ and https://silcsbio.com.

## Abstract

Antibodies bind foreign antigens with high affinity and specificity leading to their neutralization and/or clearance by the immune system. The conserved N-glycan on IgG has significant impact on antibody effector function, with the endoglycosidases of *Streptococcus pyogenes* deglycosylating the IgG to evade the immune system, a process catalyzed by the endoglycosidase EndoS2. Studies have shown that two of the four domains of EndoS2, the carbohydrate binding module (CBM) and the glycoside hydrolase (GH) domain are critical for catalytic activity. To yield structural insights into contributions of the CBM and the GH domains as well as the overall flexibility of EndoS2 to the proteins' catalytic activity, models of EndoS2-Fc complexes were generated through enhanced-sampling molecular-dynamics (MD) simulations and site-identification by ligand competitive saturation (SILCS) docking followed by reconstruction and multi-microsecond MD simulations. Modeling results predict that EndoS2 initially interacts with the IgG through its CBM followed by interactions with the GH yielding catalytically competent states. These may involve the CBM and GH of EndoS2 simultaneously interacting with either the same Fc CH2/CH3 domain or individually with the two Fc CH2/CH3 domains, with EndoS2 predicted to assume closed conformations in the former case and open conformations in the latter. Apo EndoS2 is predicted to sample both the open and closed states, suggesting that either complex can directly form following initial IgG-EndoS2 encounter. Interactions of the CBM and GH domains with the IgG are predicted to occur through both its glycan and protein regions. Simulations also predict that the Fc glycan can directly transfer from the CBM to the GH, facilitating formation of catalytically competent complexes and how the 734 to 751 loop on the CBM can facilitate extraction of the glycan away from the Fc CH2/CH3 domain. The predicted models are compared and consistent with Hydrogen/Deuterium Exchange data. In addition, the complex models are consistent with the high specificity of EndoS2 for the glycans on IgG supporting the validity of the predicted models.

Simulation input, topology, parameter and coordinate files are available from https://mackerell.umaryland.edu/software.shtml#endo_s2_data.

**Funding:** ADM, Jr. received support under grant GM131710 from the National Institutes of Health. EJS received support under grant AI132766 from the National Institutes of Health The funders had no role in study design, data collection and analysis, decision to publish, or preparation of the manuscript.

**Competing interests:** I have read the journal's policy and the authors of this manuscript have the following competing interests. ADM Jr. is cofounder and CSO of SilcsBio LLC.

## Author summary

The pathogen *Streptococcus pyogenes* uses the endoglycosidases S and S2 to cleave the glycans on the Fc portion of IgG antibodies, leading to a decreased cytotoxicity of the antibodies, thereby evading the host immune response. To identify potential structures of the complex of EndoS2 with IgG that could lead to the catalytic hydrolysis of the IgG glycan, molecular modeling and molecular dynamics simulations were applied. The resulting structural models predict that EndoS2 initially interacts through its carbohydrate binding module (CBM) with the IgG with subsequent interactions with the catalytic glycoside hydrolase (GH) domain yielding stable complexes. In the modeled complexes the CBM and the GH interact either simultaneously with the same Fc CH2/CH3 domain or with the two individual Fc CH2/CH3 domains separately to yield potentially catalytically competent species. In addition, apo EndoS2 is shown to assume both open and closed conformations allowing it to directly form either type of complex from which deglycosylation of either mono- or diglycosylated IgG species may occur.

## Introduction

Antibodies are naturally occurring proteins produced by B-cells with the primary task to identify, neutralize or tag antigens for removal, including viruses, bacteria, fungi and parasites.[1–5] They are composed of 4 polypeptide chains; two heavy chains and two light chains that adopt immunoglobulin folds and are stabilized by extensive disulfide bonds. Antibodies are generally described in terms of the antigen binding fragment (Fab) which contains the complementarity-determining regions involved in antigen binding and a crystallizable fragment (Fc) which interacts and recruits other factors and cells to mediate immune response such as Fcγ receptors.[6, 7] Antibody-linked dysfunction can cause autoimmune diseases where the immune system recognizes parts of the body, including cells or organs, as foreign thereby inducing an immune response.[8, 9] Beyond their biological function, antibodies have also revolutionized diagnosis and therapy towards the identification and treatment of a variety of diseases, particularly in cancer therapy.[10, 11] As such, monoclonal antibodies and their derivatives constitute the largest class of therapeutic proteins under development with antibody engineering playing a critical part in the process.[12] This involves the modification of antibody sequences and/or structures to optimize their activity and/or bioavailability allowing antibodies to be used as therapeutic drugs with high efficacy and low toxicity.[13]

Central to an antibody's function, the Fc region contains a conserved glycosylation site. In immunoglobulin G antibodies (IgG), a conserved glycosylation site at Asn 297 has been directly implicated in modulating critical effector functions. For instance, the removal of the Asn-297 glycan was shown to abrogate Fcγ receptor binding and modulate effector functions such as antibody-dependent cellular toxicity.[14–17] Depending on the recombinant expression system, the glycan may be a complex type, hybrid type or high-mannose type.[18, 19] Moreover, Fc glycosylation provides stabilization of the Fc structure and provides an important heterogeneity of protein glycoforms.[20, 21]

Microbes and pathogens have evolved strategies to take advantage of the conserved Asn-297 glycan to evade immune response. The pathogen *Streptococcus pyogenes* secretes enzymes called Endoglycosidase S (EndoS) and S2 (EndoS2) that catalyze the hydrolysis of the conserved glycan from antibodies thereby allowing the pathogen to evade the immune system. [22–24]. These enzymes are endo-β-N-acetylglucosaminidases that specifically catalyzes the hydrolysis of the β-1,4 linkage between the first two N-acetylglucosamine residues of the

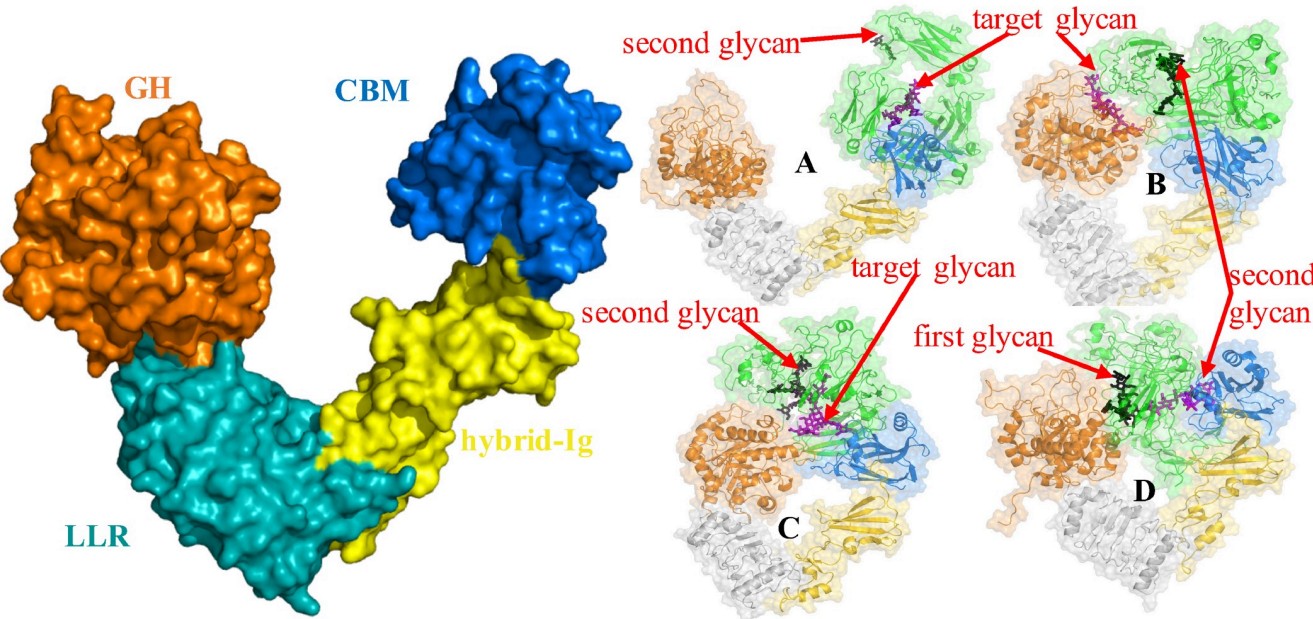

**Fig 1. Left Panel: Structure of EndoS2 with annotated domains.** Surface representation showing the general fold of EndoS2, including the glycoside hydrolase (GH; orange), leucine-rich repeat (LRR: cyan), hybrid IgG (hybrid-IgG: yellow) and carbohydrate binding module (CBM: blue) domains. Right panel (A to D): Initial conformations of Fc-EndoS2 complexes. Glycan locations are indicated with red arrows with models A to D shown on panels A to D, respectively. Only in the case of model D, which requires both glycans to be interacting with EndoS2, were both glycans simultaneous considered during model building while for the other three model only one of glycan, indicated as "target glycan" was used for model building.

biantennary complex-type N-linked glycans of IgG Fc regions.[23, 24] Specific to IgG antibodies, EndoS and EndoS2 display different glycoform specificity. Recent studies have reported the structure EndoS2 in the absence (PDB ID 6E58) or presence (PDB ID 6MDS and 6MDV) of glycan in the active site and elucidated structural determinants within the active site to explain the glycoform selectivity.[25] The structures reveal that EndoS2 is a monomeric "V-shaped" protein, composed of four different domains from N- to C-terminus: (i) a glycoside hydrolase (GH) domain (residues 43–386) that contains the catalytic site; (ii) a leucine-rich repeat (LRR) domain (residues 387–547); (iii) a hybrid-Ig domain (residues 548–680); and (iv) a carbohydrate-binding module (CBM; residues, 681–843), as shown on Fig 1.[25]

The EndoS2 GH domain adopts an $(\alpha/\beta)_8$-barrel topology typical of GH18 family enzymes that cleave glycosidic bonds. The GH domain contains several loops that connect the α-helices and β-strands, shaping a long cavity in which the *N*-glycans bind, including β1−β2 (loop 1; residues 72–102), β2−α2 (loop 2; residues 106–115), β3−α3 (loop 3; residues 140–158), β4−α4 (loop 4; residues 185–195), β5−α5 (loop 5; residues 227–235), β6−α6 (loop 6; residues 250–261), β7−α7 (loop 7; residues 285–318), and β8−α8 (loop 8; residues 339–375).[25, 26] EndoS2 recognizes complex type (CT) biantennary glycans (bisected and non-bisected), high mannose (HM) and hybrid glycans, with a preference for CT glycans over HM glycans.[26] The binding site has excellent shape complementarity for a biantennary carbohydrate, forming two distinct grooves to accommodate each of the antennae; Groove 1 is formed mainly by loops 2, 3 and 4, and binds the α(1,6) antenna. Groove 2 is formed mainly by loops 1, 2, 7 and 8, and binds the α(1,3) antenna. Loop 2 bisects the two grooves with an aromatic residue. The catalytic residues lie more proximal to the β-barrel, on the $\beta_4$-strand and loop. Loop 6 contributes a conserved glutamine Q250 and tyrosine Y252 to the active site. [25, 26]

CBMs are catalytically inactive domains frequently found on enzymes that modify carbohydrates with four main roles: glycan proximity, targeting, disruption and adhesion. The binding site of CBMs is generally defined by the location of aromatic, solvent-exposed side chains composed of tryptophan, tyrosine and occasionally phenylalanine residues. In EndoS2, single point mutation of W712 on the CBM resulted in a loss of activity. Swapping both the GH and CBM from EndoS2 onto EndoS produce an enzymatic activity nearing wild-type EndoS2, while swapping each domain individually leads to a significant loss of activity, indicating the EndoS2 GH domain and CBM coevolved to optimally recognize its substrate repertoire.[25] Further, hydrogen-deuterium exchange mass-spectrometry (HDX-MS) experiments with IgG and on a catalytically inactive mutant of EndoS2 have yielded additional insights about the role of the CBM. The HDX-MS result reported a large decrease in deuterium uptake in GH in the presence of IgG consistent with glycan binding at the catalytic site.[25] However, HDX-MS also reported the largest decrease in uptake in the CBM of EndoS2 with extensive protection in regions surrounding W712. In contrast, mild protection is observed on the bound antibody with some effects on the Fc, particularly in regions adjacent to the glycan.[25] Altogether, these observations raise the question of the structural role of the CBM in the catalytic cycle of EndoS2. Although the HDX-MS is a strong indication that the CBM engages in interactions with the antibody, it remains unclear to what extent the CBM interacts with the glycan, the protein region of the Fc or both and how those interactions can lead to a catalytically competent complex.

In this work, state-of-the-art computational tools are applied to provide an atomistic-detailed picture of different states of the EndoS2-Fc complex *in silico*. The approach applied involves individual MD simulations of the mono and diglycosylated Fc, including enhanced sampling Hamiltonian Replica Exchange, and of apo EndoS2 from which an ensemble of conformations of each system are obtained. SILCS (site-identification by ligand competitive saturation)[27–29] simulations were performed on the individual CBM and GH domains from which functional group affinity patterns, termed FragMaps, were obtained. A reconstruction process was then undertaken in which the SILCS FragMaps were used for docking of the Fc-glycan onto either or both the CBM and GH of EndoS2. From this procedure four possible models of the EndoS2-Fc complex were generated and subjected to multi-microsecond unrestrained MD simulations in order to obtain structural and dynamical properties of the complexes. From these analyses, insights into the roles of the CBM, GH and overall structure of EndoS2 in the catalytic cycle are obtained, including insights into the mechanism by which EndoS2 is highly specific for glycans on IgG.

## Results

While experimental structures of EndoS2 are available there are no structures of the full EndoS2-Fc-glycan complex thus far. Obtaining such complex structures, modeled or experimentally determined, would be particularly valuable in characterizing potential interactions of the Fc with the CBM, GH or both. Mechanistically, the interpretation of these interactions would be made in the context of both mono or diglycosylated Fc as both species must be populated throughout a catalytic cycle that results in a fully deglycosylated Fc. Accordingly, structures of the complex need to account for the possibility of the Fc being either mono or diglycosylated and consider all potential interactions of the CBM and GH with the Fc glycan or protein CH2/CH3 domains or both. To address this challenge a reconstruction approach, as described in detail below, was undertaken that took advantage of enhanced sampling HREST-bpCMAP and multi-microsecond MD simulations to generate ensembles of conformations of the individual components of the complex. From these simulations root-mean-

square difference (RMSD) clustering was used to select conformations which were then used in SILCS-Monte Carlo (SILCS-MC) glycan-protein docking from which 60,000 full complex models were generated that represent a wide range of distances between N297 and the CBM or GH glycan binding sites (S1 Fig of the supporting information). Culling of those 60,000 conformations based on extreme steric overlap yielded a total of 10,750 conformations with further culling performed based on energetic criteria in conjunction with the loop modeling. Loop modeling exploited the availability of 100 conformations of selected loops in the GH of EndoS2 allowing for reconstruction of the complexes in which minimal steric overlap in the modeled EndoS2-Fc complex occurred. This was essential to obtain conformations that could be successfully relaxed using a combination of energy minimization and restrained MD simulations from which stable structures were obtained for multi-microsecond MD simulations. From this process a total of 2000 EndoS2-Fc complex models were selected for further analysis.

Analysis of the reconstructed conformations was undertaken to identify conformations representative of the following 4 classes of models (Table 1). Model A is defined based on the distance between the predicted glycan binding pocket on the CBM and a selected glycan of the Fc to investigate the potential stability of the interactions of the CBM with the Fc glycan and adjacent CH2/CH3 domain. Model B is defined as a selected Fc glycan occupying the GH active site without additional contacts of the second glycan with EndoS2. This class of models was selected to investigate the stability of interactions of the glycan directly with the active site of the GH. Model C is defined based on both the CBM and GH simultaneously coming into contact with the Fc in the vicinity of the same glycan on either the mono or diglycosylated Fc. Such a model probes the stability of simultaneous interactions of the CBM and the GH with the same Fc CH2/CH3 domain. Model D is defined based on the CBM and GH interacting simultaneously in the vicinity of each respective individual glycans on the diglycosylated Fc, to probe if the presence of simultaneous interactions with the individual Fc CH2/CH3 domain are stable. The criteria used to select representative conformations for the different models are shown in Table 1 along with the goal for investigating each model with the values associated with those criteria shown in Table 2.

Conformer selection first identified complexes that meet the criteria for models C and D as these require simultaneous interactions of both the CBM and GH with the Fc glycan(s). Subsequent conformer selection from the remaining structures identified those that met the criteria of models A and B. Based on conformations that yielded the best agreement with the criteria

**Table 1. Initial model selection criteria and goal of each of the four EndoS2-Fc-glycan models.**

| Models | Selection criteria | Goal | Comments |
|--------|--------------------|------|----------|
| Model A | • Distance between COM of CBM binding pocket and Fc-glycan 6-Arm | • Stable interaction of the Fc with the CBM | • Both mono and diglycosylated species considered. |
| Model B | • Distance between GH active site COM and N297 of Fc | • Stable interactions of the Fc-glycan with the GH active site | • Crystal structure of glycan bound to GH exists<br>• Diglycosylated species considered |
| Model C | • Distance between glycan COM and COM of CBM binding pocket and<br>• Distance between GH active site COM with N297 of Fc associated with the same glycan | • Stable interactions of the same Fc with both the CBM and GH<br>• Can the CBM assist in positioning the glycan in the vicinity of the GH active site? | • Both mono and diglycosylated species considered |
| Model D | • Distance between GH active site COM and N297 of Fc and<br>• Distance between COM of the second glycan and COM of CBM binding pocket | • Stable interactions of the individual CH2/CH3 domain in the diglycosylated Fc-glycan each with the CBM and GH<br>• Can the CBM assist in positioning the glycan in the vicinity of the GH active site | • Diglycosylated species considered |

**Table 2. Distance between N297 and the CBM or GH in the initial models.** For model C the same N297 was used for both CBM and GH, while for model D the N297 from one chain was used for the CBM and the N297 from the second chain was used for the GH. N297 corresponds to the Fc glycosylation site. For models A and B, the values in parentheses correspond to the distance to the second N297. For model A, first glycan was selected based on the glycan interacting with CBM and for model B, first glycan was selected based on the glycan interacting with GH.

| | Model_A | | Model_B | | Model_C | | Model_D | |
|---|---|---|---|---|---|---|---|---|
| | Distance between N297 and Center-of-Mass of selected CBM and GH residues | | | | | | | |
| RUN | CBM (*) | GH (*) | CBM (*) | GH (*) | CBM (*) | GH (**) | CBM (*) | GH (**) |
| 1 | 6.74 (38.98) | 70.0 (64.68) | 30.56 (47.44) | 12.98 (40.17) | 13.56 | 12.27 | 15.22 | 11.47 |
| 2 | 16.71 (48.91) | 68.54 (61.83) | 17.33 (35.59) | 17.06 (40.22) | 19.82 | 8.41 | 15.33 | 8.87 |
| 3 | 7.34 (38.11) | 69.64 (61.11) | 40.59 (17.46) | 15.23 (58.89) | 20.57 | 13.94 | 20.91 | 25.14 |
| 4 | 9.43 (38.15) | 64.59 (61.14) | 23.20 (52.14) | 13.59 (45.62) | 21.70 | 10.72 | 12.47 | 7.34 |

* Residue 708, 710, 712, 743, 745, 748, 812, 820 and 822

** Residue 184, 186, 250 and 252

in Table 1 (*eg.* those with the shortest minimum distances specified for each model) the top 20 conformations were selected for each model. Interaction energy analysis between the glycan and the CBM or GH followed by visual inspection of the top 20 conformations was then undertaken from which 4 conformers for each model were selected for multi-microsecond MD simulations. Representative conformations of each model are included in Fig 1A—D. Full details of the molecular modeling approach are presented in the computational methods section.

Four multi-microsecond MD simulations of each model were performed for durations ranging from 2 to 5.5 μs. These simulations were performed to check the overall stability of the different complexes. In addition, conformational changes during the MD simulations were observed that yielded insights into structural transition that may contribute to deglycosylation of both mono and diglycosylated Fc by EndoS2. In addition, four 2 μs MD simulations of apo EndoS2 were analyzed to determine the range of accessible conformations that may contribute to formation of the full EndoS2-Fc complex.

Analysis of the multi-microsecond MD simulations focused on characterizing conformations associated with four selected models (Table 1) and of apo EndoS2 alone. Analysis of the EndoS2-Fc complex simulations was conducted by monitoring the contact area of the CBM or GH with the glycans or the Fc protein CH2/CH3 domains as a function of simulation time along with the associated probability distributions. Additionally, the overall conformation of EndoS2 was analyzed in the complexes as well as in apo EndoS2. Conformations of the complexes and of the apo protein from the MD simulations were then interpreted with respect to the catalytic cycle of EndoS2, specifically to provide insights about the roles of the CBM and GH in stabilizing the EndoS2-Fc complexes that would allow for deglycosylation.

## Apo EndoS2

Initial analysis focused on the range of conformations sampled in the apo-EndoS2 MD simulation. This involved analysis on the sampling of open versus closed states based on the relative orientation of the CBM and GH domains. Structurally, this sampling was monitored using the distance between the domains, the overall V-shape of the EndoS2 protein (Fig 1), as previously presented [25], and the relative rotation of the CBM to the adjacent hybrid-IgG domain. Details of the distances and angles describing the EndoS2 conformation are shown in S2A–S2D Fig. Presented in Fig 2A are time courses of the distance between the CBM and GH domain from the four 2μs MD simulations. As is evident, significant conformational variation occurred in the apo protein. Based on the distance between the domains, it is clear that open

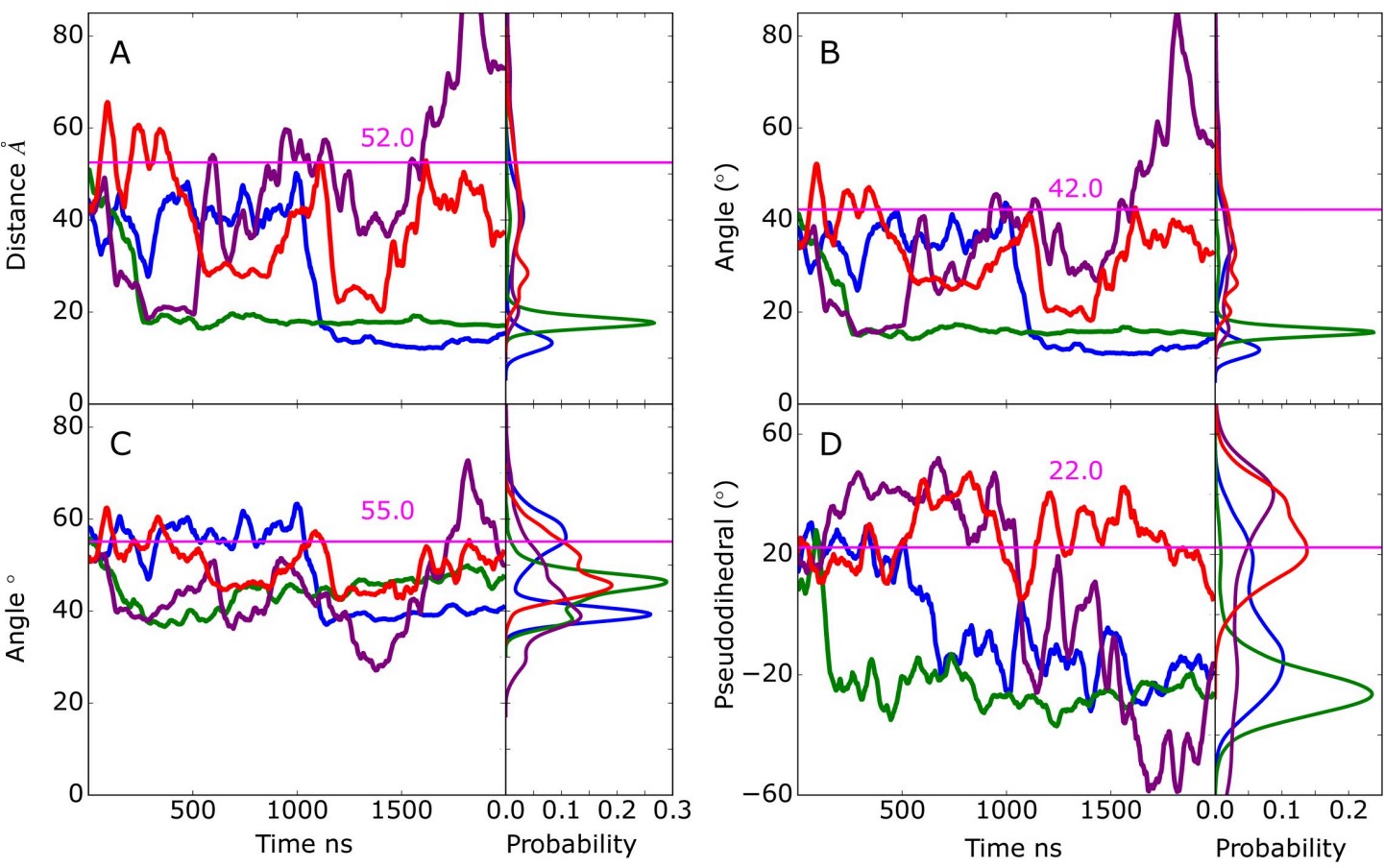

**Fig 2.** Time series and probability distributions from the apo-EndoS2 MD simulations of the A) distance between the Cα atoms of residues 186 and 712, the angles between the Cα atoms of residues B) 186-548-712 and C) 186-548-639 and D) the pseudodihedral angle defined by residues 106-508-808-712 as shown in S2A–S2D Fig. Results shown for individual MD simulations 1) blue, 2) green, 3) purple and 4) red. Data is presented as a 50 ns moving average. Values from the EndoS2 crystal structure for the respective terms are shown as horizontal lines along with the measured values in pink.

states consistent with the crystallographic distance of 52 Å (PDB ID 6E58) are being sampled, as well as closed states in which the distance between the CBM and GH domains is much smaller, down to less than 20 Å. To better understand the nature of this motion, two angles were analyzed. The first is based on the overall EndoS2 structure using residues in the GH, LRR and CBM domains (Fig 2B) and the second is based on residues in the GH, LRR and hybrid IgG domains, thereby excluding contribution of rotational motion of the CBM to the V-shape angle (Fig 2C). The overall V-shape angle in Fig 2B shows significant variations including values equivalent to or greater than that occurring in the crystal structure as well as significantly smaller angles corresponding to the shorter distances in Fig 2A. The range of angles sampled was from ~10 to ~80˚ throughout the 4 simulations. However, when the CBM is omitted from the definition of the V-shape angle, there are significantly smaller variations (~30 to 70˚) indicating the motion of the CBM relative to the hybrid IgG domain contributes to the sampling of the open versus closed states. The nature of this motion was investigated by analyzing a pseudodihedral angle defining the relative orientation of CBM to the GH based on the Cα atoms of residue 712 (CBM), 808 (CBM), 508 (LLR) and 106 (GH) (S2 Fig). The corresponding time series in Fig 2D shows that the variation in this angle is also correlated with the CBM to GH distances in Fig 2A, with more positive values corresponding to closed states.

Pearson's correlation coefficients of the CBM to GH distance with the full V-shape angle, omit CBM V-shape angle and the pseudodihedral angle are 0.988, 0.614 and -0.462 indicating that both the variations of the overall V-shape EndoS2 and the rotation of the CBM contribute to the sampling of the closed versus open states. Supporting the contribution of the rotation of the CBM to sampling of the two states is a 2-dimensional distance versus pseudodihedral angle plot shown in S2E Fig. These results illustrate the range in global conformations that apo EndoS2 can sample with both open states, as seen in crystallographic studies, as well as closed states where CBM and GH move significantly closer to each other. The sampling of this range of conformations in the apo simulation allowed for successful building of the four models presented in Table 1 as detailed in the following sections.

### EndoS2/Fc complexes

Subsequent analysis focused on the investigating the nature and stability of interactions in the respective modeled complexes in μs time scale MD simulations as follows.

### Model A

Model A was developed to probe the stability of the interactions of the CBM with the Fc. The time course of the contact area between the CBM or GH and the glycan(s) or the protein CH2/CH3 domain(s) of the Fc are shown in Fig 3. As is evident in Fig 3A, the contact area between the SILCS-predicted CBM glycan binding site (S3 Fig) and the selected glycan is approximately constant through the four simulations, with values fluctuating between 400 and 600 Å$^2$. Significant contacts between the CBM and the CH2/CH3 protein domains of the Fc that host the selected glycan are also observed and stable throughout the simulations, sampling values between 400 and 800 Å$^2$ (Fig 3C). During all the simulations the contact area between the second glycan and the GH maintained small values with the exception of a short period in simulation 1 (Fig 3B). However, the contact area between the GH with the same Fc CH2/CH3 protein domain that host the selected glycan dramatically increased after approximately 500 ns in all four simulations as shown in Fig 3D. Thus, the extended simulations predict conformations in which the interactions of the CBM with the Fc through both the glycan and protein of the Fc are stable on the time scale of the simulations. In addition, EndoS2 is predicted to simultaneously assume conformations that also allow the GH to interact with the Fc, with those interactions involving the same protein CH2/CH3 domain of the Fc with which the CBM is binding.

The initial and final frames of Fc-EndoS2 complex from model A simulation 2 are shown in S4A Fig. It includes conformations of the Fc from the initial and final frames following alignment with respect to EndoS2. In the initial conformation there is a significant interaction solely between the CBM and Fc while at the end of the simulation the Fc is also in contact with the GH. The ability of EndoS2 to simultaneously contact the same Fc CH2/CH3 domain through both the GH and CBM is associated with a conformational change which involves the rotation of the CBM allowing it to remain in contact with the first glycan and its respective CH2/CH3 chain of the Fc while bringing the Fc into contact with the GH (final location of CBM omitted in S4A Fig for clarity). Of note, in this conformation EndoS2 largely maintains an overall open conformation although some closing occurs during the trajectory (see below).

### Model B

The goal of model B was to investigate if a glycan within the context of an Fc would remain stably bound to the GH active site as required for catalysis, irrespective of other contacts. In all 4 model B MD simulations, the selected glycan is initially localized in the GH active site based

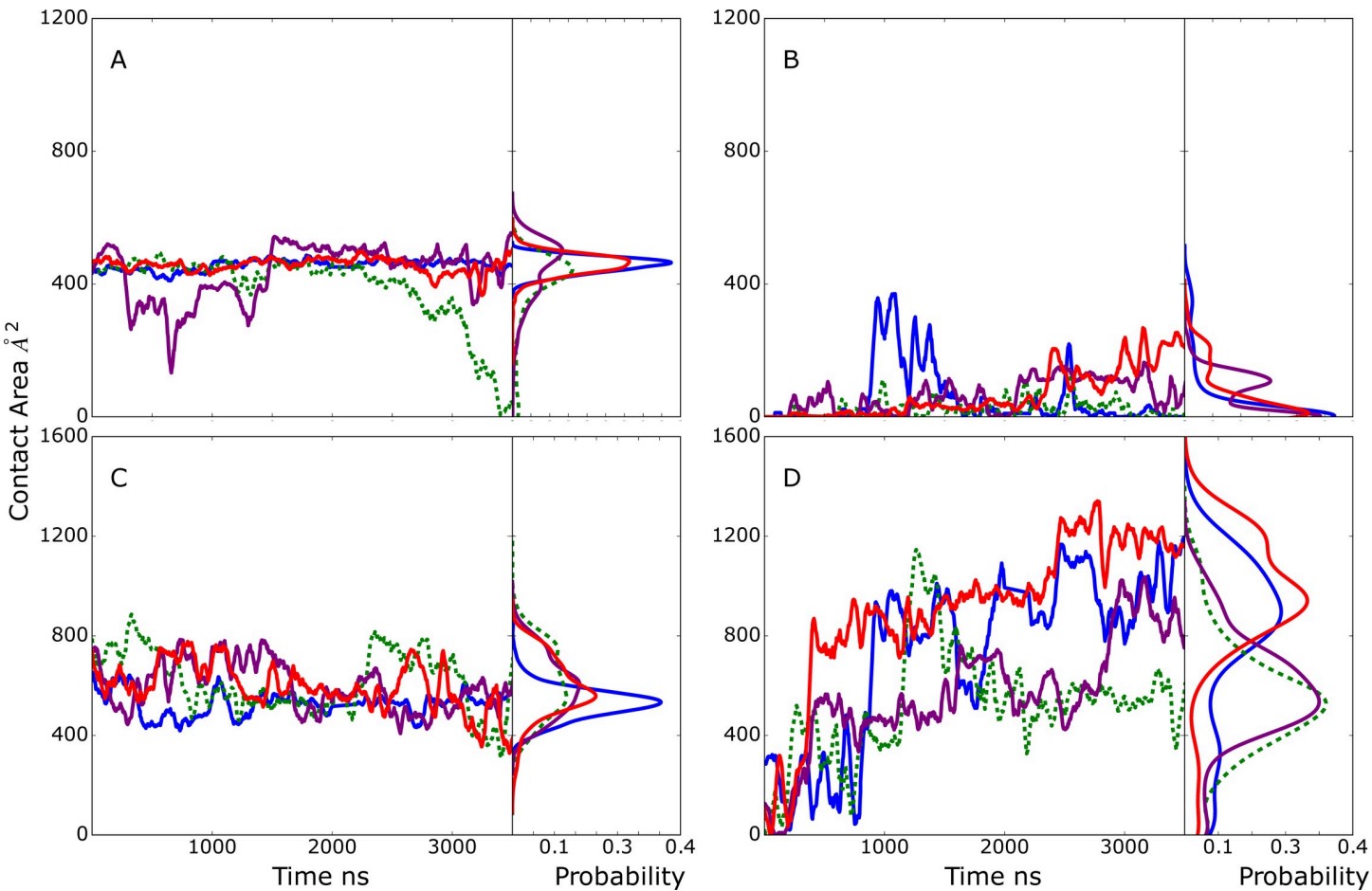

**Fig 3.** Model A time course and probability distributions of the contact area, Å$^2$, between A) the selected glycan of the Fc and the CBM, B) the second glycan of the Fc and the GH, C) the Fc CH2/CH3 protein domains hosting the selected glycan and the CBM and D) the Fc CH2/CH3 domains hosting the selected glycan and the GH during the MD simulations. Results shown for individual MD simulations 1) blue, 2) green, 3) purple and 4) red. Data is presented as a 50 ns moving average. Dashed lines indicate that simulation 2 corresponds to the monoglycosylated Fc with the GH (panel B) interacting with the first monosaccharide and fucose of the second glycan. The remaining simulation systems are diglycosylated.

on the SILCS modeling. As all four simulations proceed the extent of glycan-GH contacts decreased. In three of the simulations, these contact area values consistently decreased below the 684 Å$^2$ observed in the glycan-bound crystallographic structure with the extent of interactions approaching zero in the case of simulation 3 (Fig 4B). Interestingly, in three out of four simulations, limited contacts between the CBM and the selected or the second glycan in the system are formed and maintained (Fig 4A) while large contact areas of the CBM with the Fc protein CH2/CH3 domain that hosts the selected glycan are present (Fig 4C). Interactions of the GH with the protein CH2/CH3 domain are also present, though they are generally less than that with the CBM, with the exception of simulation 3 in which the contacts with the glycan are almost totally lost (Fig 4B) while substantial contacts with the Fc protein CH2/CH3 domain occur (Fig 4D). These results predict that it is unlikely that EndoS2 relies solely on interactions between GH and the Fc-glycan to achieve a stable, catalytically active complex.

The properties of simulation 2 (Fig 4, green lines) are particularly interesting. In this simulation, the contacts between the GH and the selected glycan are maintained to the largest extent as are the interactions of the CBM with the protein CH2/CH3 domain that hosts the

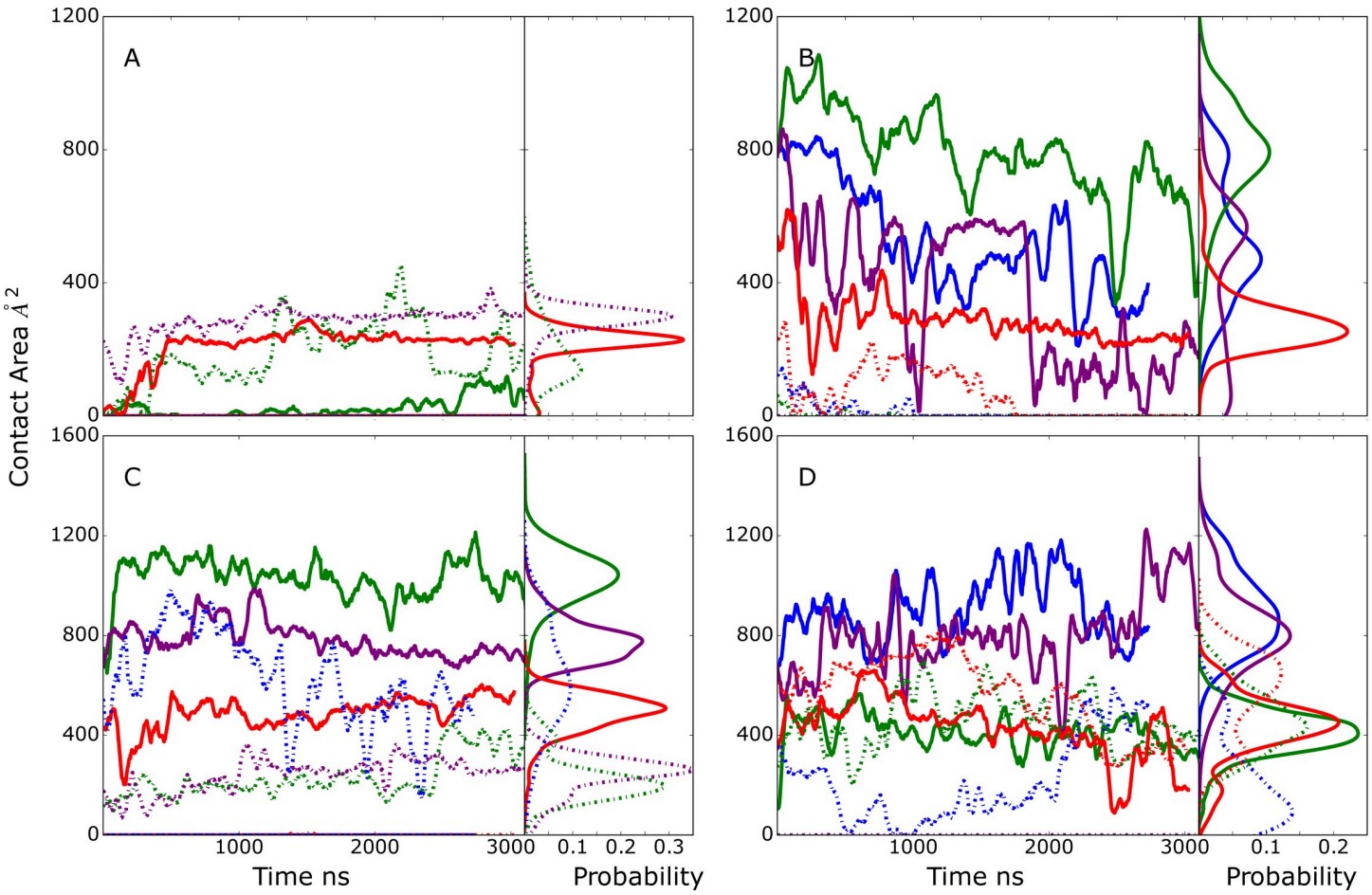

**Fig 4.** Model B time course and probability distributions of the contact area, Å², between A) one of the Fc glycans (see below) and the CBM, B) the selected glycan of the Fc and the GH, C) the Fc protein CH2/CH3 domain hosting the selected glycan and the CBM and D) the Fc protein CH2/CH3 domain hosting the selected glycan and the GH during the MD simulations. Dark-solid lines represent selected glycan (A and B) or the Fc protein CH2/CH3 domain that hosts the selected glycan (panel C and D) and light-dashed lines represent second glycan that was not selected to interact with the GH (A and B) or the Fc protein CH2/CH3 domain that hosts the second glycan (C and D). Results shown for individual MD simulations 1) blue, 2) green, 3) purple and 4) red. Data is presented as a 50 ns moving average.

selected glycan. Indeed, there is an increase of ~400 Å² in the CBM-protein contacts at the beginning of that simulation. This behavior would suggest that interactions of the CBM with the CH2/CH3 domain that hosts the glycan to which the GH binds leads to stabilization of the GH-glycan interactions. Note that this scenario is similar to that predicted to occur in model C (see below) but contrasts model D where interactions of the CBM with the second Fc glycan or associated CH2/CH3 domain are predicted to stabilize the interactions of the GH with the first glycan.

The initial and 2000 ns frames for model B simulation 2 are shown on S4B Fig. The Fc is in an orientation that allows the CBM to interact with the same protein CH2/CH3 domain of the Fc that hosts the glycan to which the GH is bound. Similar complexes occurred in simulations 1 and 4 though the contact of the GH with the glycan and CBM with the CH2/CH3 domain are decreased as compared to simulation 2. From simulation 2, the conformation of EndoS2 is predicted to assume a partially open conformation that allows for interactions of both the CBM and GH with the protein and glycan region of the Fc to be maintained.

## Model C

Model C was designed to identify conformations in which both the CBM and the GH active site are in the vicinity of the same Fc glycan. Initial selection criteria included the distance between N297 and the GH active site COM, allowing determination of when the glycan is adjacent to the GH active site but not necessarily occupying the site. Similarly, the distance of the same glycan from the SILCS-predicted CBM binding pocket COM was used as a second criteria to assess simultaneous proximity to the CBM. While conformations meeting these criteria were identified (Table 2), the extent of contacts of the CBM and GH with the glycan were relatively small (Fig 5A). With simulations 3 and 4, a low level of contact of the glycan with the CBM was maintained, with the amount of contact increasing in simulation 3 (Fig 5A). In contrast, significant contacts of the CBM with the Fc protein CH2/CH3 domain were observed and maintained in 3 out of the 4 simulations (Fig 5C). In the case of the GH, contacts with the glycan were observed in simulations 2 and 3 (Fig 5B) while significant contacts of the GH with the CH2/CH3 domain of the Fc were present in all the simulations (Fig 5D). While the extent of contacts of the GH with the glycan are relatively small, analysis of the distance between

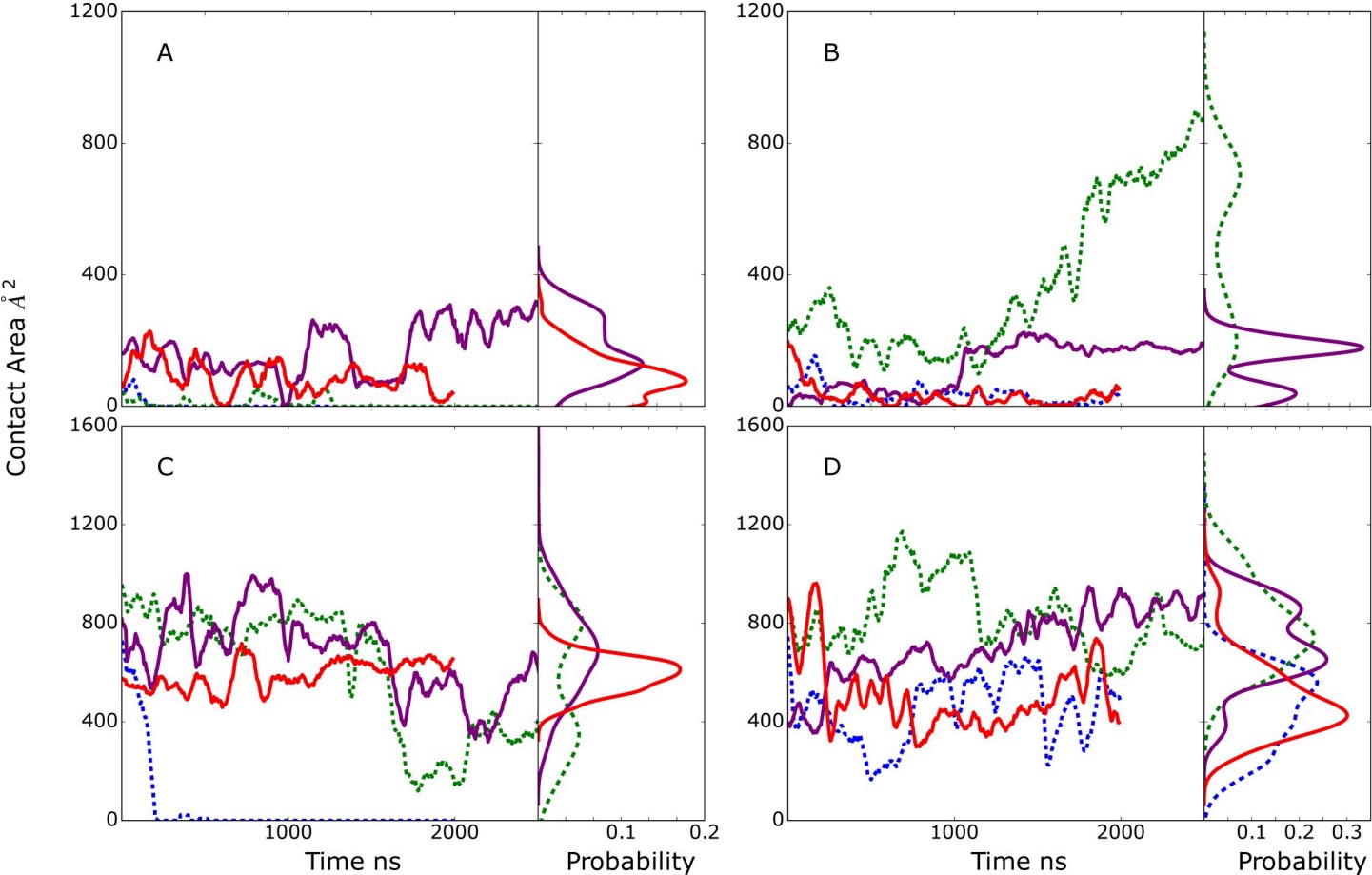

**Fig 5.** Model C time course and probability distributions of the contact area, Å$^2$, between A) the selected glycan of the Fc and the CBM, B) the selected glycan of the Fc and the GH, C) the Fc protein CH2/CH3 domains hosting the selected glycan and the CBM, and D) the Fc protein CH2/CH3 domains hosting the selected glycan and the GH during the MD simulations. Results shown for individual MD simulations 1) blue, 2) green, 3) purple and 4) red. Data is presented as a 50 ns moving average. Dashed lines indicate in simulations 1 and 2 the Fc is monoglycosylated such that the CBM (panel A) and GH (panel B) are interacting with the same glycan.

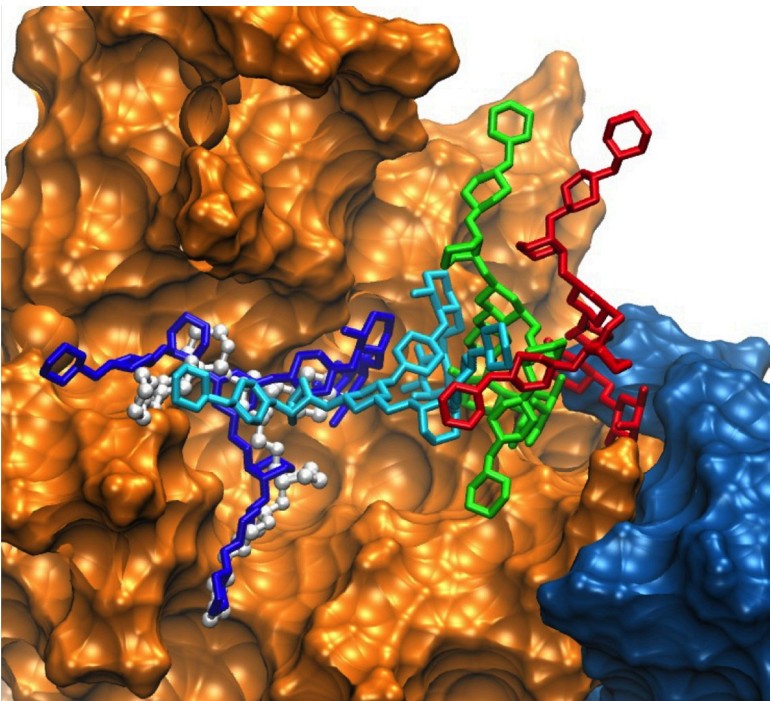

**Fig 6. Transfer of the glycan from CBM (blue surface) to GH (orange surface) from model C simulation 2.**
Different locations of the glycan in sticks are color highlighted. For the glycan red corresponds to its initial location between the GH and CBM, green corresponds to an intermediate (900 ns) state, cyan corresponds to a second intermediate (1624 ns) state and dark blue corresponds to the final state (3296 ns) with the glycan in the GH active site. Crystallographic conformation of the bound glycan is shown in white ball and stick representation. For visualization, the Fc protein chains were removed.

N297 and the COM of the GH glycan binding site were typically in the range of 10 Å in the 4 MD simulations, indicating that the glycan is in vicinity of the GH active site.

The behavior of simulations 2 and 3 were of note. In simulation 3, contacts involving the glycan and protein CH2/CH3 domain on the same Fc chain with both the CBM and GH were maintained throughout the simulation predicting that both the CBM and GH can interact simultaneously with the region around the same glycan on the Fc. In the case of simulation 2, there is a transfer of the glycan from the CBM into the GH active site as indicated by the loss of interactions of the Fc with the CBM over the course of the simulation (Fig 5A and 5C) while there is a substantial increase in the interactions of the GH with the Fc glycan (Fig 5B). Structural images showing the change in the orientation of the glycan relative to the CBM and GH are shown in Fig 6. Initially, the glycan engages in some interactions with the CBM, although most interactions of the CBM occur with the Fc CH2/CH3 domain (Fig 5A and 5C). This conformation is similar to the complex structure discussed above from the model B simulations as shown in S4B Fig. Subsequently, the glycan shifts away from the CBM and towards the GH active site consistent with the increase in the GH-glycan contact area (Fig 5B). By the end of the simulation, the glycan assumes a conformation within the GH active site which is similar to that observed in the glycan-bound EndoS2 crystal structure (compare blue stick and white CPK representation of the glycan in Fig 6) with the RMSD being 3.3 Å for all glycan non-hydrogen atoms. Thus, the simulations predict that the CBM and GH of EndoS2 can simultaneously interact with the region around the same glycan on the Fc. This result combined with the model B system predicting the Fc-glycan-GH interactions to be unstable indicates that

EndoS2 may require simultaneous interactions of both the GH and the CBM with the glycan or adjacent protein CH2/CH3 domain in the monoglycosylated Fc to allow for deglycosylation of that species. Indeed, the simulation 2 results predict that a glycan initially in contact with the CBM can shift into an orientation in the GH active site.

The initial and final frames showing the orientation of the Fc relative to EndoS2 are shown in S4C Fig for simulation 2. The overall orientation of the Fc relative to the CBM and the GH shows only local conformational changes occurring in the Fc. The conformation of EndoS2 is in the closed state, which facilitates simultaneous interactions of both the CBM and GH with the same glycan or CH2/CH3 domains on the Fc (see below).

### Model D

Model D was designed to probe conformations of the EndoS2/Fc with one glycan in the vicinity of the CBM binding site and the second in the vicinity of the GH active site. Initial selection criteria included the distance between N297 and GH active site and the COM of the second glycan with the COM of the CBM binding pocket (Table 2). This model is only possible with the diglycosylated Fc. Results in Fig 7 show the initial model D conformations were largely stable throughout the four 2 μs MD simulations. In almost all cases the interactions of the CBM and GH with both the glycan and protein chain of the Fc were maintained or increased over the course of the simulations. The only exception was simulation 1 where contacts of the GH with the glycan decrease towards the end of the simulation (Fig 7B) though contacts with the CH2/CH3 domains are maintained (Fig 7D). These results predict that the simultaneous interaction of the GH and CBM with the individual glycans and surrounding regions of the Fc can occur and maintain a stable complex, potentially corresponding to a catalytically competent complex suitable for deglycosylation of the diglycosylated species.

Initial and final orientations of the Fc relative to EndoS2 are shown in S4D Fig following alignment to EndoS2 from the initial frame of simulation 2. As is evident, the orientation of the Fc glycan relative to EndoS2 did not change significantly during the simulation, indicating its stability. In addition, EndoS2 is observed to be largely in an open conformation, which allows the Fc to be flanked on both sides by EndoS2 as simultaneous interactions with the two glycans and the GH and CBM, respectively, occur.

### EndoS2 conformational properties in the complexes

Visual inspection of the conformations of the modeled complexes show a range of open versus closed states of EndoS2 (S4 Fig). To illustrate this in more detail, the CBM to GH distances were extracted from the MD simulations performed from the four developed models along with the associated probability distributions (Fig 8). The plots include a line indicating the value observed in the crystal structure which is considered to be an open conformation (PDB ID: 6E58, CBM to GH of ~52 Å). As is evident, varying degrees of open versus closed states are sampled in the simulations. Models A and B sample a range of states, consistent with those models including individual interactions of either the CBM or GH with the Fc, such that a specific conformation of EndoS2 is not essential. With the model C simulations, the CBM-GH distance is significantly shorter than that observed in the crystal structure and is largely maintained throughout all the simulations. This is indicative of a closed conformation required for simultaneous interaction of the CBM and GH with the same glycan on the Fc, as shown in S4C Fig. In model D, the distances similar to that observed in the crystal structures are maintained throughout the course of the simulations. This is consistent with model D maintaining open conformations required for simultaneous interaction of the CBM and GH domains with regions in the vicinity of individual glycans on the diglycosylated Fc as shown in S4D Fig. The

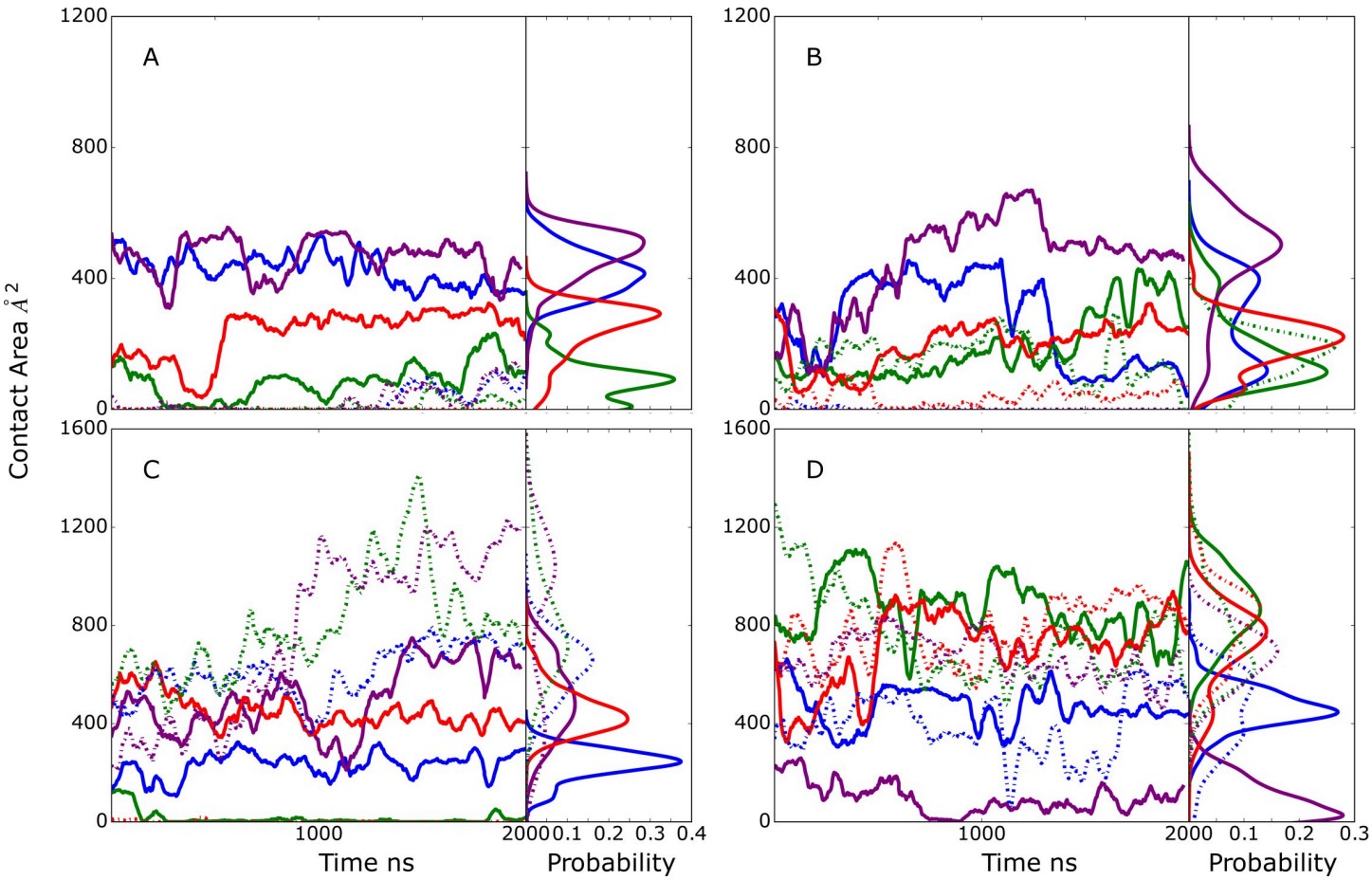

**Fig 7.** Model D time series and probability distributions of the contact area, Å², between A) the first glycan (glycan interacting with CBM) of the Fc and the CBM, B) the second glycan (interacting with GH) of the Fc and the GH, C) the Fc protein chain hosting the first glycan and the CBM, and D) the Fc protein chain hosting the second glycan and the GH during the MD simulations. Results shown for individual MD simulations 1) blue, 2) green, 3) purple and 4) red. Data is presented as a 50 ns moving average. Light-dash lines represent the contact area between second glycan or Fc CH2/CH3 domain hosting the second glycan and CBM (panel A and C) and between first glycan or Fc hosting first glycan and GH (panel B and D), while solid lines represent the contact area between first glycan or Fc CH2/CH3 domain hosting first glycan and CBM (panel A and C) and between second glycan or Fc hosting second glycan and GH (panel B and D).

stability of the CBM to GH distances in the model C and D simulations gives additional support for the stability of the C and D model structures. Notably, the range of open and closed states seen in the EndoS2-Fc complexes fall within the range sampled in the apo protein as shown above in Fig 2A. Therefore, sampling of this range of conformations by the apo protein suggests that the conformational dynamics of the EndoS2 could allow the protein to bind to the diglycosylated Fc as observed in model D or to the monoglycosylated form as observed in models A and C.

## IgG-EndoS2 modeled complexes

Given that EndoS2 interacts with full antibodies in the experimental regimen, models of the full antibody-glycan-EndoS2 complexes were generated. These models were based on the final conformations obtained from selected simulations of each of the models obtained by aligning the Fc of the antibody (PDB ID: 1IGT) with the Fc in each of the complexes. The resulting structures are shown in S5A–S5D Fig. In all cases the orientations of the Fabs from the full antibody have minimal steric overlap with EndoS2. As there is significant variability in the

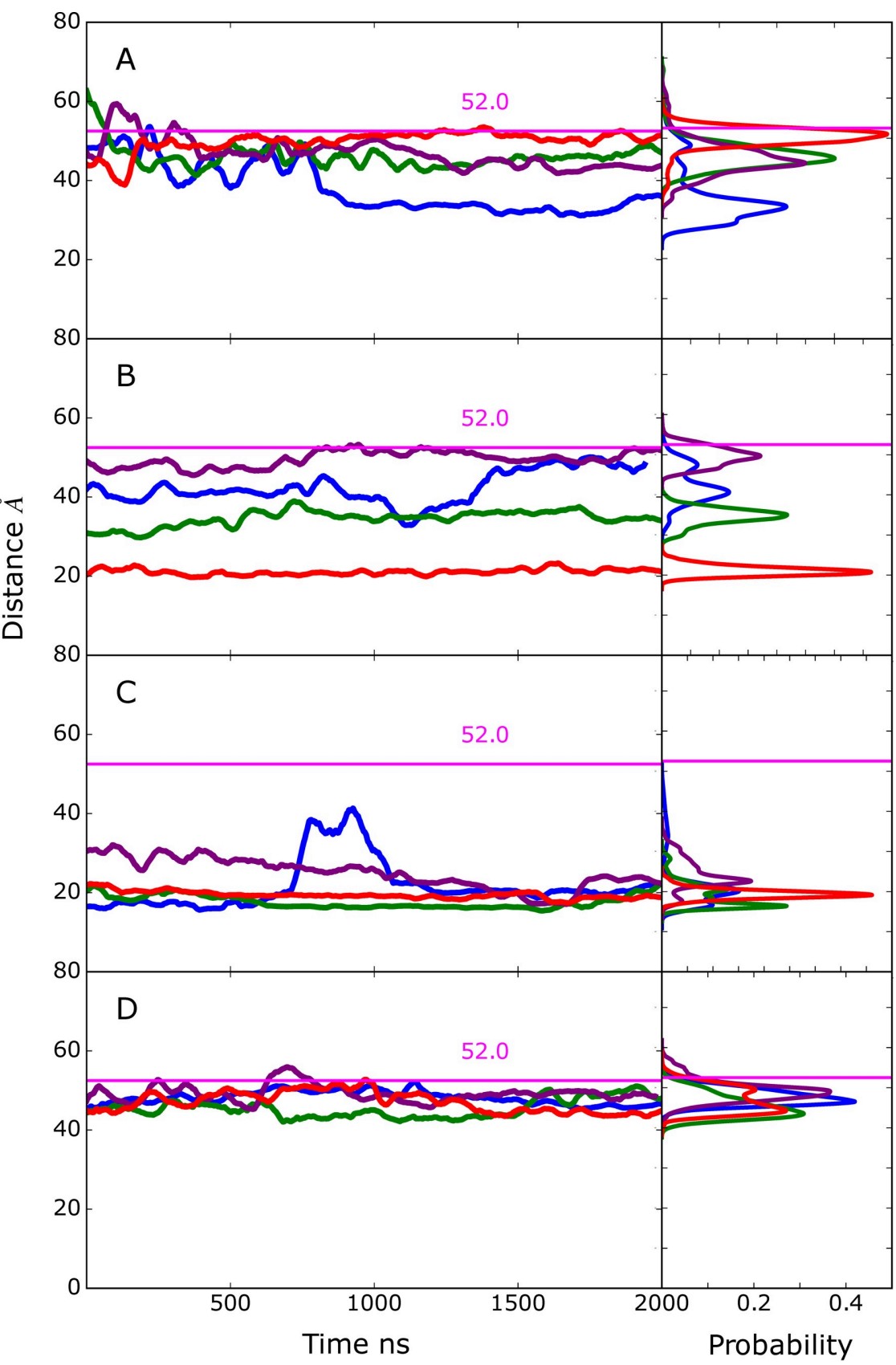

**Fig 8.** Time series and probability distributions for the CBM to GH distances based on the Cα atoms of residues 186 and 712 for the MD simulations on the A) Model A, B) model B, C) model C, and D) model D. The corresponding distance in the crystal structure (PDB ID: 6E58) is indicated with horizontal magenta line along with the measured value. Results shown for individual MD simulations 1) blue, 2) green, 3) purple and 4) red. Data is presented as a 50 ns moving average.

relative orientations of the Fc and Fabs in antibodies due to the flexibility of the linker, the extent of steric overlap should be readily avoided by conformational relaxation.[30] Accordingly, it may be assumed that the results based on the Fc-EndoS2 modeling and simulations may be extrapolated to the full IgGs.

## Comparison to hydrogen-deuterium exchange experimental data

The ensemble resulting from the modeling and multi-microsecond MD simulations were quantitatively compared with HDX experimental data [25]. To that end, the protection factors (PF) of the amide hydrogens were calculated directly from the trajectories according to Eqs 1 through 3 (see below). The difference in predicted percent deuterium uptake upon complex formation was then calculated based on the difference between apo-EndoS2 simulations and the pooled complex simulations using only the diglycosylated multi-microsecond MD simulations to be consistent with the diglycosylated experimental results. In addition, model specific average percent differences of the individual diglycosylated species are shown in S6A and S6B Fig. Fig 9A presents both experimental and predicted percent differences at 10 s of labeling and a protein surface annotated image is shown in Fig 9B. Overall, the calculated results are qualitatively in agreement with the experimental results, particularly with respect to the protection occurring primarily on the CBM and GH. Notable is the localized agreement in the protection pattern of loop 7 (residues 285–318) and part of loop 8 (residues 339–375) of the GH. Similarly, on the CBM the peptides comprised of residues 741 to 748, which is part of the SILCS-determined CBM glycan binding site, show high protection in the simulations consistent with the experimental protection data. This indicates that the modeling of both the GH- and CBM-Fc interactions is consistent with the HDX experiment. Interestingly, the greater amplitude of protection in the CBM versus the GH observed experimentally is largely recapitulated in the simulated data (Fig 9A). It is also worth noting that in some localized areas of the CBM, such as peptides 694–721, 739–740 and 811–823, the calculated PFs identify but underestimate protection. However, more quantitative agreement between the calculations and experimental data may not be anticipated given, among other reasons, the significant time scale difference of the calculated and experimental methods. Regardless, it suggests that the pooled ensemble of all models, on average, may underestimates contacts that the CBM undertakes with the Fc protein and/or glycan.

The HDX PFs and subsequent differences in percent deuterium uptake were also calculated for the Fc as an average over all models, but only in the context of diglycosylated species as was used in the experiments (Fig 10). The corresponding values for individual models using only the diglycosylated species are presented in S7 Fig and mapped onto the Fc surface in S8 Fig. In the experimental study, the percent deuterium uptake change was minimal on the Fc (Fig 11A) and significantly lower in amplitude than that occurring with EndoS2 (Fig 10A). In overall agreement, the calculations also predict reduced amplitude of protection from HDX on the Fc as compared to that on EndoS2. However, the calculated results show significantly larger protection on the Fc as compared to that observed experimentally. Analysis of the individual models suggest that the larger percent differences is due to the contributions of model D (S8 Fig, note the difference in scales in panel D vs. A, B and C). Interestingly, the calculated result for model C is in the closest quantitative agreement to the experimental HDX protection,

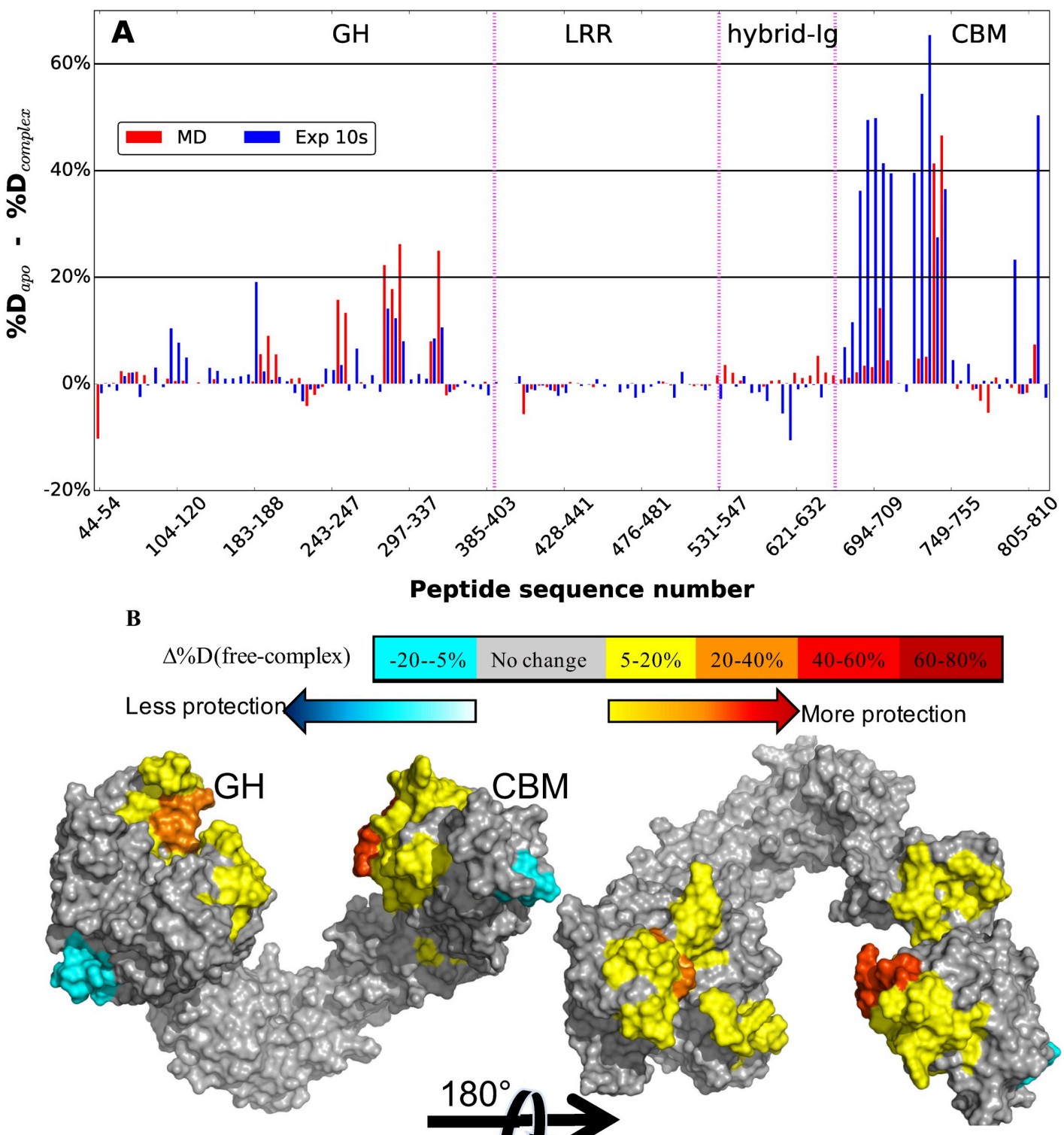

**Fig 9.** A) Percent difference in hydrogen-deuterium exchange from experimental studies [25] and from multi-microsecond MD simulations for the diglycosylated species averaged over all the models. Experimental differences in percent deuteration between peptides from EndoS2 in the apo and IgG1-complexed diglycosylated state after 10s are plotted as blue bars and the average percent difference from the apo EndoS2 and Fc-glycan-EndoS2 complex simulations as red bars. The different domains of Endos2 are labelled across the top of the figure and separated by purple dashed lines as follows: Glycoside hydrolase (GH), leucine-rich repeat (LRR), hybrid-Ig, and carbohydrate-

binding module (CBM). Individual peptides are plotted on the X-axis from the N- to C-terminus based on the sequence number of the first residue in the peptide. Some of the peptides are skipped from the label for better visualization. B) Calculated differences in hydrogen–deuterium exchange between apo Fc and Fc-glycan-EndoS2 complex mapped onto a surface representation of the EndoS2 (PDB ID 6E58).[25] Glycoside hydrolase (GH) and carbohydrate-binding module (CBM) domains are labeled.

indicating minimal contact between the Fc and both the CBM and GH of EndoS2 (S7C Fig). Given the greater agreement of the calculated difference in deuterium uptake of model C with respect to the experimental data, the results suggest that the conformations predicted based on model C may be the dominant species, though the relatively low protection observed experimentally may be due to the highly structured and stable nature of the immunoglobulin fold that may not afford as much protection upon complex formation as the loops of GH and CBM. However, together the presented calculations as well as the HDX experiments do indicate that some level of interactions between the CH2/CH3 domain of the Fc and EndoS2, particularly the CBM, are occurring.

## CBM interactions with the Fc glycan versus CH2/CH3 domain

Experimental functional studies on EndoS2 have established a number of mechanistic insights on the activity of the protein. The protein catalyzes the hydrolysis of the glycosidic linkage between the first two N-acetylglucosamine units on the target Fc glycan. The protein can act on multiple types of glycans including CT, high mannose (HM) and hybrid type glycan.[25, 26, 32] However, and notably, EndoS2 is highly specific for the glycans on IgG versus glycans bound to other proteins including other engineered sites the Fc on IgG. As discussed above, with EndoS2 there are larger levels of protection observed in HDX experiments in the region of the CBM versus the GH [25], indicating the presence of interactions of Fc protein-CBM interactions, consistent with the predicted model C and D results shown above. Consistent with this is an experimental mutation study of three solvent-exposed aromatic residues on the CBM of EndoS2, W712, Y819 and Y820 (S3D Fig), which were mutated to A712, S819 and S820, leading to a loss of activity in all cases.[25, 26, 33] The activity loss did not distinguish HM and CT glycan. To help interpret this result, the contact made by those residues with either the protein or glycan portion of the Fc was analyzed (Fig 11). Both the protein and glycan portions of the Fc were consistently in contact with these CBM residues in model A while for model B, C and D contacts were dominated by protein-protein interactions. The results support the prediction that CBM functions, at least in part, through interactions with the CH2/CH3 domain of the Fc.

## Glycan extraction

The Fc glycan undergoes extensive interactions with the CH2/CH3 domain of the Fc in the absence of other interacting molecules based on crystallographic structures.[34] Accordingly, it is necessary for the glycan to be "extracted" from the surface of the Fc CH2/CH3 domain to allow it to interact with the GH and undergo hydrolysis. The possible mechanism of glycan extraction in the complex simulations was investigated through examination of the contact area between Fc CH2/CH3 domain and the target glycan. In model A simulation 4 and model C simulation 2 extraction of the glycan was observed, with the latter leading to the glycan being transferred to the GH active site as shown above in Fig 6. As shown in Fig 12A, in model A the contact area between the Fc CH2/CH3 domain hosting the target glycan goes from upwards of ~600 Å$^2$ down to less than 100 Å$^2$ after 4500 ns, while in the case of model C (Fig 12B) the glycan contact with the Fc CH2/CH3 domain decreases shortly after 1000 ns. With model A loss of contact initially involves the 6-Arm followed by the 3-Arm while in model C

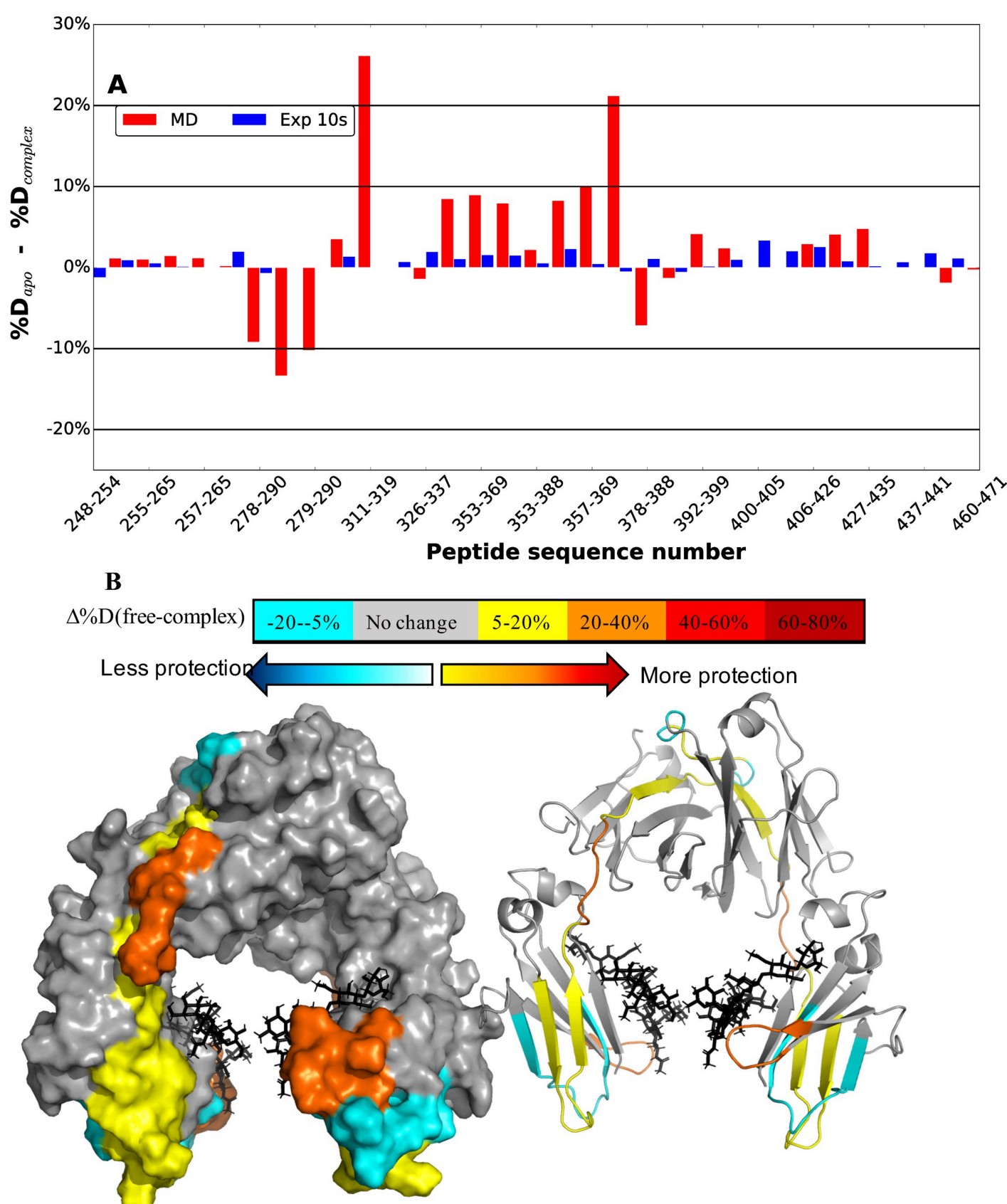

**Fig 10.** A) Average of diglycosylated model simulation vs experimental hydrogen-deuterium exchange. Differences in percent deuteration between peptides from Fc in the unliganded and IgG1-complexed state for experimental [25] after 10s are plotted as blue bars and percent difference from the apo-Fc and Fc-glycan-EndoS2 complex simulations as red bar. Individual peptides are plotted on the x-axis from the N- to C-terminus based on the sequence number of the first residue in the peptide. Some of the peptides are skipped from the label for better visualization. B) Calculated differences in hydrogen–deuterium exchange between apo-Fc and Fc-glycan-EndoS2 complex mapped onto a surface representation of the Fc (PDB ID 1IGT). [31] Crystal bound glycan are labeled in black stick.

both Arms shift away from the Fc CH2/CH3 domain at the same time. Interestingly, in both cases shortly after the decrease in the contact area between the glycan and the Fc CH2/CH3 domain, the CBM loop comprised of residues 734–751 extends, assuming a location previously occupied by the target glycan. In the case of model C, the glycan comes into contact with the second, non-host Fc CH2/CH3 domain, corresponding to the movement of the glycan into the GH active site discussed above. Images of the target glycan and the CBM 734–751 loop before and after extraction in model A in Fig 12C shows how the loop occupies the region on the surface of the Fc CH2/CH3 domain previously occupied by the glycan. Such an orientation is consistent with the high protection factor of residues 741–748 observed in the HDX experiments, as discussed above. Thus, two of the simulations indicate that extraction of the glycan from contact with the Fc CH2/CH3 domain is facilitated by CBM loop 734–751, which moves

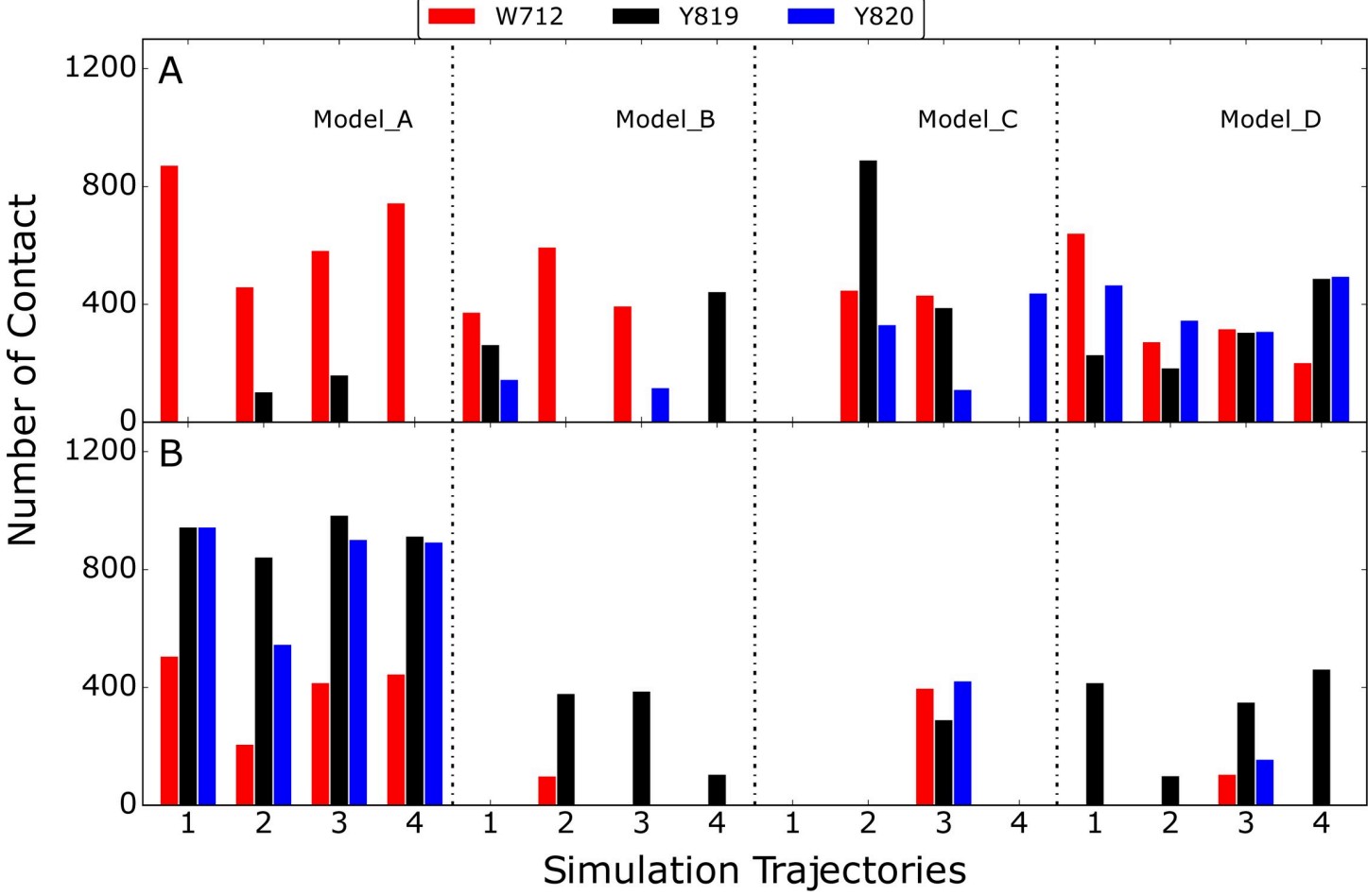

**Fig 11.** Number contact between W712, Y819 and Y820 on the CBM and A) protein portion of Fc or B) glycan portion of Fc. Results are shown for the four models with the four individual bar graphs under each model corresponding to four independent MD simulations. Number of contacts was calculated by counting frames with heavy atom that are within 4 Å from those selected amino acids based on snapshots extracted every 50 ns from the simulations.

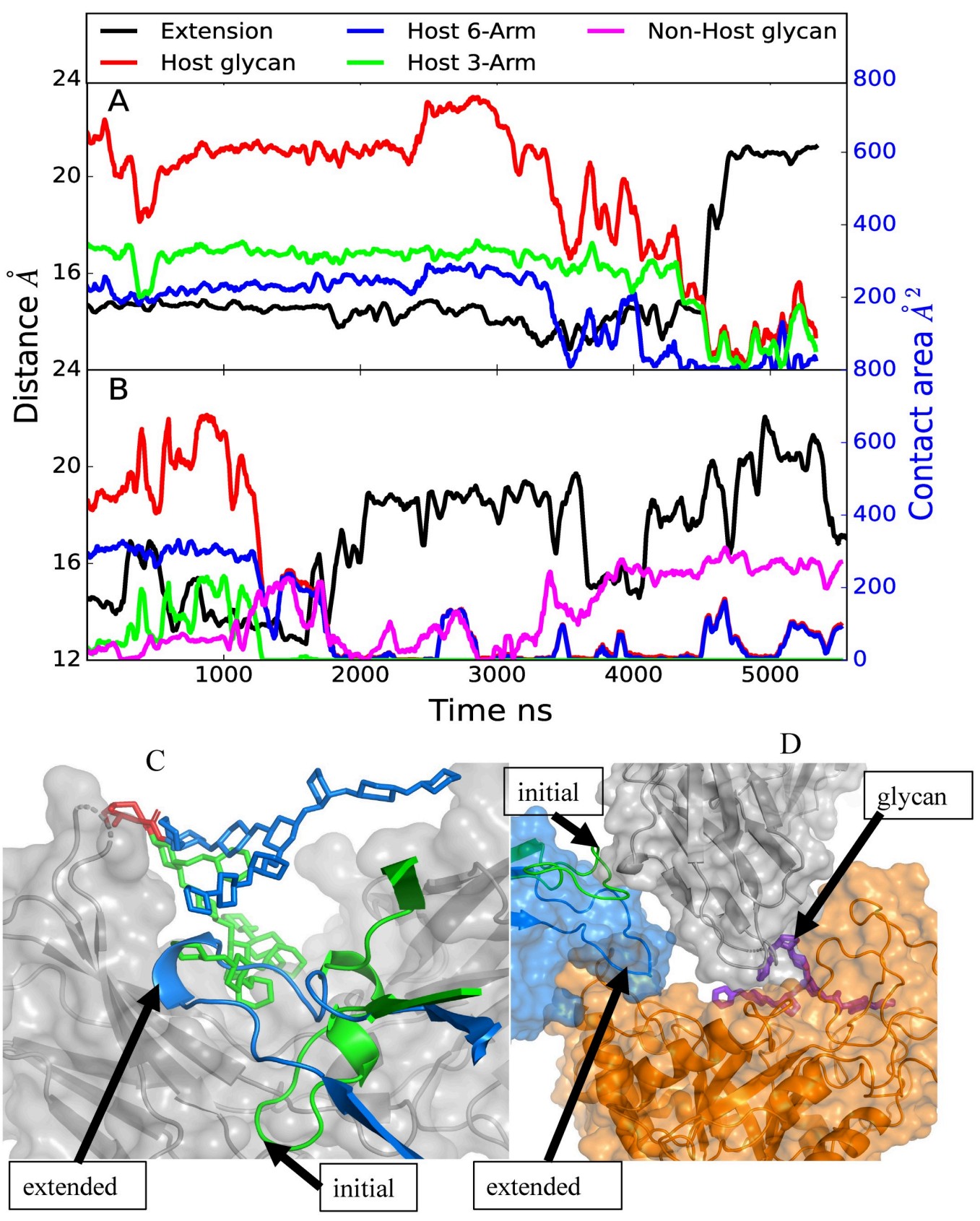

**Fig 12.** A) Model A and B) model C time course of the extension of CBM loop 734–751 (black, Å) and contact area, Å$^2$, between the target glycan and the Fc CH2/CH3 domain hosting the target glycan (red), between target glycan and the Fc CH2/CH3 domain hosting the second glycan (violet), between 6-Arm of the target glycan and the Fc CH2/CH3 domain hosting the target glycan (blue) and between 3-Arm of the target glycan and the Fc CH2/CH3 domain hosting the target glycan (green). The extension distance is measured from the middle of the loop based on residue 741 and one of the end of the loop based on residue 734 and plotted along left Y-axis in black and contact area is plotted along the right Y2-axis with blue ticks. Results from trajectory 4 for model A and trajectory 2 for model C are shown. Data is presented as a 50 ns moving average. Images of the C) model A and D) model C initial and final conformations of the CBM loop. For model A, the initial glycan (green stick), CBM loop (green cartoon), and final glycan (blue stick) and CBM loop (blue cartoon) are shown with the Fc CH2/CH3 in the background (gray surface). Shown for model C is the Fc CH2 domain (gray surface overlaid on cartoon representation), GH (orange surface overlaid on cartoon representation), and CBM (blue surface) with the initial CBM (green cartoon) and final (blue cartoon) conformations of the 734–751 loop. The locations of the loop are indicated by the labeled black arrows.

under the glycan potentially blocking the glycan from re-engaging the Fc CH2/CH3 domain. In the case of model C this allowed the glycan to move into the GH binding site, as shown on Fig 6, with the extended loop providing support for the Fc CH2 hosting the glycan that slide into the GH active site.

## Discussion

The results from the model building and multi-microsecond MD simulations allowed probing the conformational landscape of the EndoS2/Fc complex and for prediction of the complex structures between an IgG and EndoS2 that may participate in the catalytic cycle. Initial interactions between the antibody and EndoS2 are predicted to occur primarily through the CBM based on the model A simulations, with initial interactions with the GH unlikely based on the model B simulations. The individual multi-microsecond simulations of over 2 to 5.5 μs of each system showed the model A complex to be stable over the course of the 4 simulations, including interactions with both the glycan and CH2/CH3 regions of the Fc. The importance of the antibody-CBM interactions is consistent with the experimental data discussed above showing the total loss of activity of EndoS2 due to the mutation of 3 aromatic residues in the CBM.[25, 26, 33] The model B simulations, where interactions of the GH with the glycan significantly decrease, predict that interactions solely involving the GH with the Fc glycan are not stable, predicting the need for additional interactions of the CBM with the IgG to attain catalytically competent complexes.

The model C simulations yield insights into possible simultaneous interactions of both the CBM and the GH with the same Fc protein chain. States where the CBM and GH are interacting with the same Fc chain involve conformations of the protein that are more closed than observed in the crystallographic structures (Fig 8). The initial models and subsequent simulations predict that such simultaneous interactions can occur, though the contacts largely involve interactions with the CH2/CH3 regions of the Fc (Fig 5). Interestingly, a similar complex was predicted to occur in model B simulation 2 further supporting the validity of this complex structure. In addition, the structural changes in model C simulation 2 suggest that the glycan can transfer from a region adjacent to the CBM to the GH active site (Fig 6). This observation suggests that glycan recognition by the CBM may play a role in guiding the glycan into the GH active site. Under such a scenario, the CBM would first interact with the glycan and progressively transfer it to the GH while increasing contacts with the CH2/CH3 portions of the Fc.

Model D was designed to predict the feasibility of the individual glycans and surrounding regions of two Fc chains simultaneously interact with CBM and GH domains, respectively. Such a complex would imply that the CBM would primarily act as an anchor, interacting with the other Fc chain, glycan and/or CH2/CH3, while the GH hydrolyzes the first glycan. All 4 model D simulations showed the complexes to be stable. However, the glycan did not interact with GH in orientations similar to that observed in the crystal structure based on RMSD of the

simulated glycans being ~25 Å or greater throughout the simulations. Such interactions are predicted to correspond to a conformation that may potentially lead to a catalytically competent complex allowing for initial deglycosylation of the diglycosylated species. Once hydrolysis occurs the antibody could either fully dissociate from EndoS2 or remain in contact through interactions of the Fc the CBM potentially leading to a model C type complex.

The predicted simultaneous interactions of the CBM and the GH required for stable EndoS2-IgG interactions are suggested to be relevant to the known specificity of EndoS2 for glycans on N297 of IgGs.[25] While hydrolysis of glycans alone are known to occur, the protein does not deglycosylate other proteins or IgG with glycosylation at other residues efficiently.[23, 25] This suggests that to attain a catalytically competent complex some level of specificity of the interactions of the CBM with the Fc must occur simultaneously with GH-Fc glycan interactions required for catalysis. These could occur both in the model C and model D scenarios where the specific interactions occur with the same Fc CH2/CH3 domains as that being deglycosylated or with the second Fc CH2/CH3 domains that may occur in the diglycosylated IgG, respectively. The low-level of activity of EndoS2 against individual glycans appears to be associated with transient interactions of the glycan with the GH, consistent with the ability of the enzyme to catalyze the reverse "glycosylation" reaction.[23, 25] Furthermore those mutations on CBM that affected IgG hydrolysis had no impact on the AGP hydrolysis of the glycans on $\alpha_1$-acid glycoprotein.[25] The lack of activity with other protein-glycan systems is suggested to be due to local steric effects that destabilizes these transient interactions. These reported results are consistent with the instability of the GH-glycan interaction in model B, and suggests that the inability of EndoS2 to bind to individual glycans will limit non-specific deglycosylation thereby maximizing deglycosylation of IgG and bacterial survival.

Finally, the conformational flexibility of apo EndoS2 is consistent with both the model C and D complex structures. By being able to sample both the closed and open conformations, it is conceivable for EndoS2 to interact directly with either a mono- or diglycosylated IgG. For instance, one possible scenario would involve initial interactions occurring between the CBM of EndoS2, following which interactions with the GH occur. In the case of the diglycosylated species the open form of EndoS2 would allow for interactions of the GH with either of the glycans leading to a catalytically competent complex. Alternatively, in the case of monoglycosylated Fc the closed form of EndoS2 may allow for the GH to interact with the same Fc chain; such a complex could also occur with a diglycosylated IgG. Interestingly, the present simulations also indicate a mechanism whereby the 734–751 CBM loops facilitates extraction of the glycan from the surface of the Fc CH2/CH3 domain, allowing the glycan to bind to the GH active site as required for catalysis.

Molecular modeling, though the SILCS-based macromolecular docking and a detailed reconstruction process, yielded four initial models of possible Fc-EndoS2 complexes. Subsequent multi-microsecond MD simulations, encompassing over 50.2 μs of simulation time, allowed for predictions of stable complexes and structural transitions that may contribute to the catalytic cycle of EndoS2. While the presented models are largely consistent with published experimental data, future experimental structural studies will be essential to fully clarify the catalytic mechanism of EndoS2.

## Computational methods

Modeling and simulations were performed with the program CHARMM [35–37] using the CHARMM36 additive force field for carbohydrates [38, 39], CHARMM36m for the protein [40] and the CHARMM TIP3P [41] water model. Initial coordinates and simulation boxes were prepared with the CHARMM-GUI glycan builder [42–44] based on the crystal

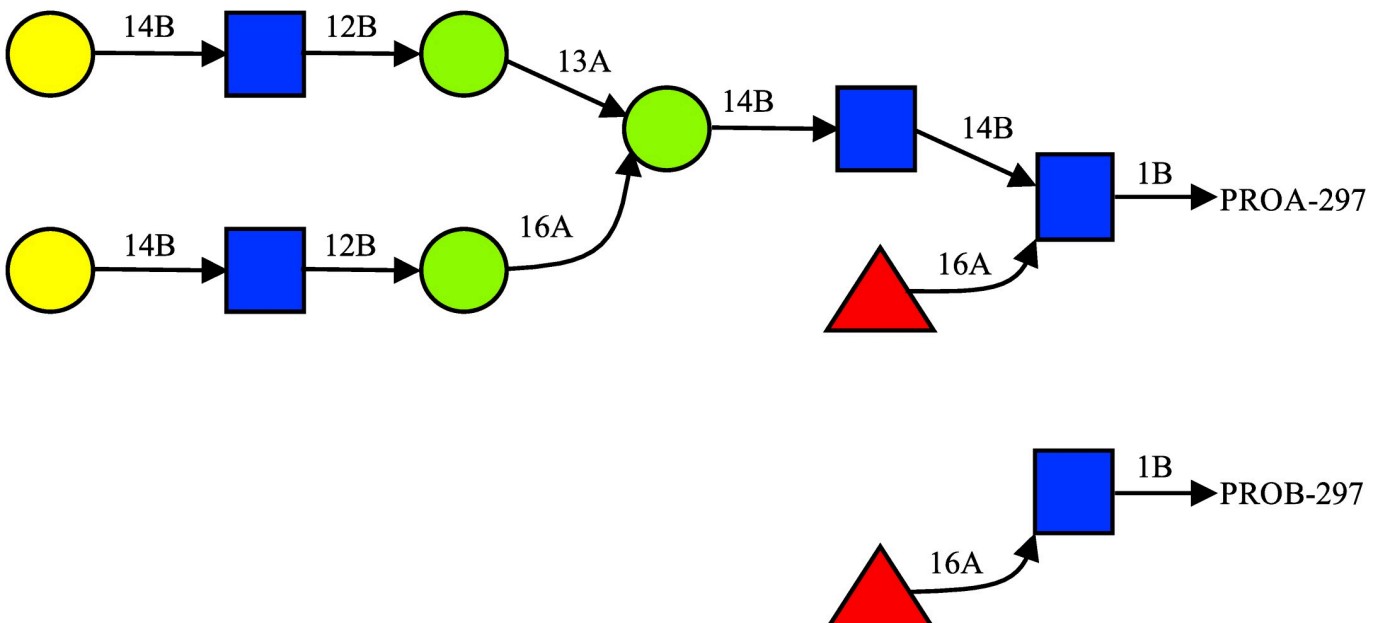

**Fig 13. Complex Type (CT) glycan used in this study.** In the case of monoglycosylated Fc, at the second glycosylation site the first monosaccharide and the attached Fucose were maintained to model the digested glycan as shown.

coordinates in protein databank (PDB) files 1IGT for the Fc and 6E58 for EndoS2 (chain A, crystal waters and ions retained).[25, 31] For the isolated glycan simulations (Fig 12), a single glycan was extracted from the IgG crystal structure (PDB ID: 1IGT [31]). Antibody 1IGT has heavy chains with chain ID B and D, light chains with chain ID A and C, and the two glycans with chain ID E and F attached to heavy chains B and D, respectively. The missing galactose on the 13A linkage arm was added using CHARMM-GUI glycan builder to generate the full CT glycan. The Fc was extracted from the IgG by removing the two Fabs at the first disulfide bond position, with the two terminal Cys residues linked through a disulfide bond. Initial Fc models were built in monoglycosylated or diglycosylated states. For monoglycosylated, the glycan on the second chain (Chain F) was removed beyond the first 13 linkage such that only the first monosaccharide and the linked fucose were maintained. The two Fc chains are identified as chain A and B with the associated glycans identified as glycan A and glycan B. We also used the following names for the two arms or antennae of the glycan as 3-Arm and 6-Arm for the glycan arms with 13 linkage and 16 linkage respectively, as shown on Fig 13.[45]

Once initial coordinates were generated for each system, they were immersed into a pre-equilibrated water box. The size of the water box was selected to extend at least 10 Å beyond the non-hydrogen atoms of the proteins or glycans. Water molecules with the oxygen within a distance of 2.8 Å of the non-hydrogen protein or glycan atoms were deleted. Information on the simulation systems is reported in S1 Table. All subsequent minimizations and MD simulations were performed in the presence of periodic boundary conditions. MD simulations were performed with CHARMM and OpenMM.[35, 36, 46]

Equilibration of the solvated systems was initiated with a 500-step steepest descent (SD) minimization followed by a 500-step adopted-basis Newton-Raphson (ABNR) minimization in which mass-weighted harmonic restraints of 1.0 kcal/mol/Å$^2$ were applied on the non-hydrogen atoms of protein and glycan. In all simulations under the NVT or NPT ensembles, including the subsequent HREST-bpCMAP simulations, the temperature was maintained at

298 K using the Hoover algorithm with a thermal piston mass of 1000 kcal/mol·ps$^2$.[47] A constant pressure of 1 ATM was maintained using the Langevin piston algorithm with a collision frequency of 20 ps$^{-1}$ and piston mass of 1630 amu in CHARMM.[48] The size of the periodic box was adjusted based on Monte Carlo Barostat to maintain constant pressure in OpenMM. [49] The covalent bonds involving hydrogen atoms were constrained with the SHAKE algorithm and a time step of 2 fs was used.[50] In the energy and force evaluations, the nonbonded Lennard-Jones interactions were computed with a cutoff of 12 Å with a force switching function applied over the range from 10 to 12 Å. The electrostatic interactions were treated by the particle mesh Ewald method with a real space cutoff of 12 Å, a charge grid of 1 Å, a kappa of 0.34, and the 6-th order spline function for mesh interpolation.[51]

Conformational sampling of the monoglycosylated Fc and diglycosylated Fc was enhanced by applying the HREST-bpCMAP (Hamiltonian replica exchange with solute tempering 2-biasing potential 2D dihedral conformational maps) method that involves concurrent solute scaling and biasing potentials.[52] Enhanced sampling was achieved by subjecting the glycosidic linkages to potential biasing with concurrent effective temperature scaling of the intra-solute potential and solute-environment interactions. All the production HREST-bpCMAP simulations were carried out in CHARMM using the replica exchange module REPDST, with BLOCK to scale the solute–solute and solute–solvent interactions [41, 53, 54] and with specific bpCMAPs applied as the 2D biasing potentials along selected glycosidic linkages.[55] The bpCMAP [52] biasing potential is applied to the glycosidic and glycopeptide linkages. The 2-dimensional grid-based bpCMAP were constructed using the corresponding disaccharide models for the glycosidic linkages and using a dipeptide connected to a monosaccharide for the glycopeptide linkage in the gas phase as described previously.[52, 56, 57] BpCMAPs were applied along the φ/ϕ dihedrals for each glycosidic linkage in the glycans and the glycopeptide linkage. A total of 8 replicas were simulated for each system and exchanges attempted every 1000 MD steps according to the Metropolis criterion. In the HREST-bpCMAP simulations, the solute scaling temperatures were assigned to 298 K, 308 K, 322 K, 336 K, 352 K, 370 K, 386 K and 405 K, with the ground-state replica temperature of 298 K selected to correspond to the experimental studies. The distribution of scaling factors for the bpCMAPs across the 8 perturbed replicas was determined as previously described and the acceptance ratio between different neighboring replicas was examined to guarantee that sufficient exchanges were being obtained.[52, 56, 58, 59] The HREST-bpCMAP simulations were run for 250 ns. Final results are presented based on the full 250 ns of sampling obtained in the ground state, 298 K replica. In addition to the HREST-BP simulations, standard MD simulations were performed using OpenMM [46] for the apo-EndoS2 system. Four independent MD simulations of 2 μs were run for a total of 8 μs to obtain adequate sampling.

## SILCS simulations of Fc, GH and CBM

Site identification by ligand competitive saturation (SILCS) simulations and analysis were performed using the SilcsBio software package (SilcsBio LLC)[28, 29] along with the GROMACS [60] simulation program. The SILCS calculations were performed on the deglycosylated Fc (PDBID: 1IGT[31]), the CBM of EndoS2 (PDBID: 6e58[25], residues 681–843) and the GH domain of EndoS2 (PDBID: 6e58[25], residues 43–386). The SILCS simulation setup included protein, water, and eight probe solutes including benzene, propane, methanol, formamide, acetaldehyde, imidazole, methylammonium, and acetate. Protonation states of ionizable residues were determined by GROMACS tools.[60] For each system, ten independent SILCS simulations initiated with randomly positioned solutes at approximately 0.25 M each were performed for better convergence. The SILCS simulations used an iterative oscillating excess

chemical potential Grand Canonical Monte Carlo (GCMC) and molecular-dynamics (MD) protocol, in which GCMC involves the sampling of water and solutes with the subsequent MD simulations allowing for sampling of protein-conformational dynamics as well as that of the solutes and water, as previously described.[27–29] Each simulation was run for 100 ns for a total of 1 μs of MD sampling. SILCS FragMaps were then generated by binning selected solute atoms into voxels of a 1 Å spaced grid spanning the simulation system. 3D normalized probability distributions were obtained by normalizing the voxel occupancies computed in the presence of the protein by the respective concentrations of the solutes based on the ratio of the total number of solutes to that of water (55 M). The normalized distributions were Boltzmann-transformed to yield grid free energies (GFE) for each FragMap, termed GFE FragMaps. All SILCS calculations used the CHARMM36m[40] protein force field with the CHARMM TIP3P [41] water model and the CHARMM general force field (CGenFF)[61–63] for the solute molecules.

## Complex reconstruction and scoring

Models of the full EndoS2 in complex with the mono and diglycosylated Fc were built based on conformations selected from the Fc-glycan and CBM-glycan (see below) HREST-bpCMAP simulations and the apo-EndoS2 MD simulations. These components were then combined through a "reconstruction" process to create a total of 6 complexes. These include 1) monoglycosylated Fc-EndoS2$_{CBM}$, 2) monoglycosylated Fc-EndoS2$_{GH}$, 3) diglycosylated$^A$ Fc-EndoS2$_{CBM}$, 4) diglycosylated$^A$ Fc-EndoS2$_{GH}$, 5) diglycosylated$^B$ Fc-EndoS2$_{CBM}$, and 6) diglycosylated$^B$ Fc-EndoS2$_{GH}$, where the superscripts A or B indicate the glycan on diglycosylated Fc being docked onto the CBM or GH and the subscripts CBM and GH indicate the domain of EndoS2 being docked to the glycan and subsequently used to position the full protein. The 6 complexes were ultimately assigned to four models of the full complex, as described below. The reconstruction process involved the following steps, which are shown diagrammatically in Fig 14.

1. For all the generated Fc-glycan conformations, the glycan was docked onto the A) CBM or B) GH using SILCS-MC (see below) using the tripeptide-glycan model conformers: (2 proteins (CBM or GH) x 3 glycans (monoglycosylated, diglycosylated$^A$ and diglycosylated$^B$) x 100 conformations = 600 conformations).

2. Reconstruct the full Fc by alignment with the tripeptide of the docked tripeptide-glycan model conformers yielding CBM-Fc and GH-Fc complexes.

3. Create full EndoS2-Fc models by aligning full EndoS2 conformations to either the CBM or GH structures (600 x 100 EndoS2 conformations = 60,000 conformations, with 10,000 conformations for each of the 6 complexes).

4. Evaluate nonbond energies to remove models with significant steric overlap of EndoS2 with the Fc (*eg*. When Fc-glycan is docked to the GH is steric overlap with the CBM occurring?).

5. Cluster remaining conformations based on distance criteria and assign to models A, B, C, or D as described below.

6. For EndoS2 conformations with secondary contacts with either GH or CBM identify respective loop conformations from the EndoS2 simulations that have minimal steric overlap with the Fc-glycan. This accounts for the local flexibility of the loops in GH or CBM.

7. Subject selected models to energy minimization and structural relaxation prior to full MD simulations.

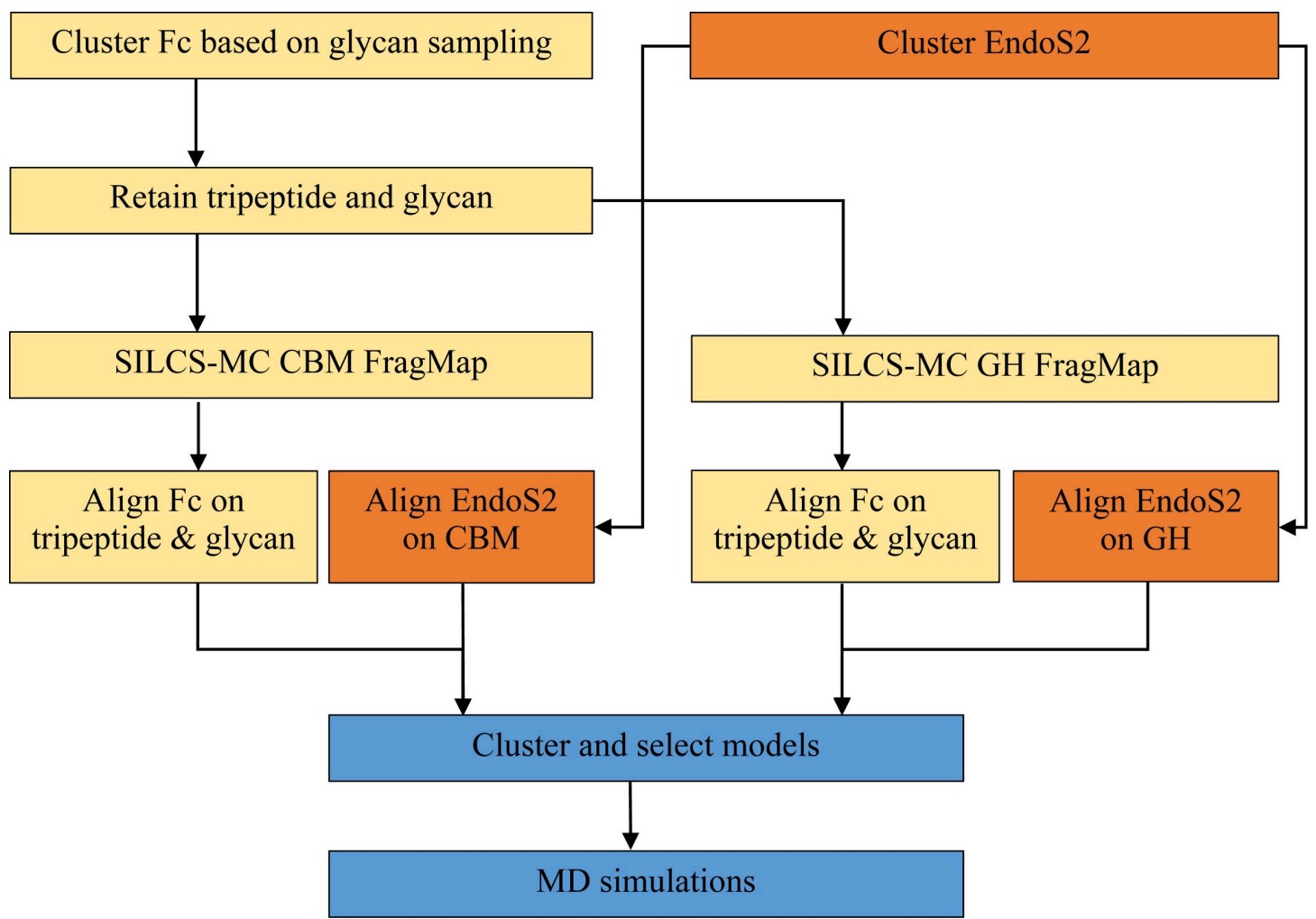

**Fig 14. Workflow used for the reconstruction of the EndoS2-Fc-glycan complexes.**

To initiate the reconstruction process representative coordinates were selected from the mono and diglycosylated Fc HREST-bpCMAP and apo-EndoS2 MD simulations. From the 75,000 and 200,000 snapshots obtained from the HREST-bpCMAP and MD simulations, respectively, 100 conformations were selected for glycan A from the monoglycosylated Fc simulation, 100 for glycan A and 100 for glycan B from the diglycosylated Fc simulation and 100 from the apo-EndoS2 simulation using RMSD clustering from the MMSTB toolset.[64] From the Fc-glycan conformations the glycan and the tripeptide centered on N297 were extracted. This was performed to facilitate docking of the glycans onto the CBM or GH followed by reconstruction of the full Fc based on alignment with the tripeptide. N297 is used as the identifier in this work to be consistent with other published studies though the residue ID in the PDB is N314.[31, 65]

The glycosylated Fc tripeptide conformations were then aligned onto either the CBM or the GH using SILCS-MC docking as described below. This was followed by reconstructing the full Fc by alignment with the tripeptide. Then the full EndoS2-Fc complex was built based on the orientation of the docked CBM or GH. This yielded the 60,000 initial models of the full

EndoS2-Fc complex. From this pool of modeled conformations, conformers corresponding to the four possible models were selected for the additional simulation and subsequent analysis.

## SILCS-MC docking of tripeptide-glycans onto CBM or GH

Docking of the tripeptide-glycans onto the CBM or GH applied the SILCS Monte Carlo (SILCS-MC) method [29] to generate and score binding poses for the glycan on the SILCS FragMaps on the proteins. In the approach MC sampling of rotational, translation and intramolecular dihedral degrees is performed using the Metropolis criteria based on the glycan CGenFF intramolecular energy and the ligand GFE (LGFE) which is based on the overlap of glycan atoms with the respective GFE FragMaps and summing those contributions, as recently described.[27] The docking poses with the most favorable LGFEs were selected.

Starting geometries for the tripeptide-glycan to CBM or GH SILCS-MC docking were obtained as follows. For GH a crystal structure with the glycan from PDB ID 6MDS[25] bound was used to define the glycan binding site and to initially pose the tripeptide glycan for SILCS-MC docking. As there is no crystal structure of a glycan bound to the CBM the binding site was identified by positioning the glycan at 15 random positions and orientations approximately 10 Å away from the protein surface using Pymol (S3A Fig) followed by SILCS-MC docking of each of the 15 orientations. The SILCS-MC docking involved three rounds of MC with the center-of-mass (COM) of the tripeptide-glycan constrained to stay within 12 Å of the initial location. The initial round involved 50,000 MC steps at 298 K with maximum translations, rotations, and dihedral rotations of 1 Å, 30° and 90°, an additional 50,000 MC steps at 298 K with maximum step sizes of 0, 0 and 90° to perform additional intramolecular conformational sampling of the glycans and a final simulated annealing MC for 100,000 steps with maximum step sizes of 0.2 Å, 9°, and 9°. Each of the 15 SILCS-MC run involved 250 individual docking runs. From these the top 15 poses based on the LGFE scores were selected for analysis from which the glycan binding site on the CBM was identified. The final docked 15 most favorable poses are shown in S3B Fig with the binding site identified based on the top 6 poses as shown in S3C Fig. Specific residues identified based on the docking along with those identified in experimental mutation studies [25] are shown in S3D Fig, with the overlap of the experimental and dock identified residues supporting SILCS-MC identification of that site.

Final generation of CBM-Fc and GH-Fc complexes involved docking the tripeptide-glycans obtained from the mono and diglycosylated Fc simulations. Step one involved manually docking one of the glycan-tripeptide glycans in the respective binding sites of the GH based on the crystal structure and the CBM in the site described in the preceding paragraph using Pymol. The remaining 299 conformations of the tripeptide-glycans were then RMSD aligned with the manually docked tripeptide-glycan using CHARMM. Each of the 300 pre-docked conformations to each of the CBM or GH were then subjected to SILCS-MC. The COM of the tripeptide-glycans were constrained to be within 20 Å of the starting orientation in the CBM or GH calculations. SILCS-MC was performed to refine the spatial location of the tripeptide-glycan in the binding site without altering the overall conformation of the glycan by performing only local relaxation of the glycan position and intramolecular conformation. The initial round of MC involved 50,000 steps at 298 K with maximum translations, rotations, and dihedral rotations of 5 Å, 3° and 5°, respectively, followed by an additional 50,000 MC steps at 298 K with maximum step sizes of 1 Å, 1° and 1°, with the final relaxation involving simulated annealing MC for 100,000 steps with maximum step sizes of 0.2 Å, 1°, and 1°. Each SILCS-MC docking involved 250 individuals MC runs. Once the tripeptide-glycans were docked into the CBM or GH the full Fc was reconstructed. This was performed by RMSD aligning the N297 tripeptide in each of the 100 Fc conformations with the respective tripeptide on the docked tripeptide-

glycan. This process yielded 300 CBM-Fc complexes and 300 GH-Fc complexes based on the CBM and GH orientations with the monoglycosylated Fc and the two glycans in the diglycosylated Fc.

Full EndoS2-Fc complexes were reconstructed using the 300 CBM-Fc complexes and 300 GH-Fc complexes and the 100 apo-EndoS2 conformations. The structures from the apo EndoS2 simulation were RMSD aligned with either the CBM or GH structures to form the complexes yielding a total of 60,000 conformations of the full EndoS2-Fc complex (2 x 300 x 100). From the 60,000 conformations of the full EndoS2-Fc complexes a distance-based matrix was generated for filtering and scoring of the complexes. Due to both computational and memory demands, temporary coordinates were not saved, instead pre-selected distances were generated. These include distances between the COM of different parts of the Fc and EndoS2, distances between N297 and the GH active site or glycan binding pocket of CBM predicted from SILCS-MC and minimum distances between terminal galactose of the glycan and CBM or GH.

During reconstruction in a number of conformations the opposite end of EndoS2 relative to that used for the Fc docking had a steric clash with the Fc protein and or its glycan (*eg*. docking performed directly to the GH could yield conformations in which the CBM in the full EndoS2 has steric clashes with the Fc). A distance-based COM calculation between different parts of the Fc and CBM or GH was used as a rapid method to initially remove overlaps followed by the calculation of the CHARMM Lennard-Jones (LJ) energy in the gas phase to remove additional conformations with a smaller number of steric clashes. Prior to the LJ energy analysis, to allow for conformational flexibility of the GH to be taken into account in the model, the conformations of loops in the glycan binding regions were sampled based on reconstruction of the loops using conformations obtained from the EndoS2 simulations. Loops were defined as follows; loop 1: residues 72–102, loop 2: residues 106–115, loop 3: residues 140–158, loop 4: residues 185–195, loop 5: residues 227–235, loop 6: residues 250–261, loop 7: residues 285–318 and loop 8: residues 339–375. For each loop a total of 100 representative conformations were obtained from the 200,000 apo-EndoS2 simulation snapshots based on RMSD clustering.[64] The loop modeling process involved swapping each of the 8 loops with the 100 conformations extracted from the simulations. The CHARMM input used for this process is included in supporting information (S1 Text) under 'CHARMM input section'.

For each of the remaining complex conformations following the initial culling based on the COM criteria, identification of unfavorable loop conformations was performed by simply deleting loops 1 through 8 individually and calculating the energy change, where steric clashes are identified as cases in which the LJ interaction energy between the loop and the remainder of the system became more favorable by 1000 kcal/mol or more. Once a problematic loop is identified that loop was then reconstructed using the pool of loop conformations from the simulations and then evaluating if the interaction energy decreased relative to the previous conformation. This was performed sequentially starting with loop 1 through loop 8 based on the assumption that loop-loop interactions were negligible. The procedure yielded conformations that were then further evaluated based on energies followed by energy minimization and MD as described below.

## EndoS2-Fc complex simulations

The selected model complexes were immersed in a pre-equilibrated cubic water box. The size of the water box was selected based on the condition that it extend at least 10 Å beyond the non-hydrogen atoms of the complex model. Water molecules with the oxygen within a distance of 2.8 Å of the non-hydrogen solute atoms were deleted. For all of the subsequent

minimizations and MD simulations, periodic boundary conditions were employed. A list of the systems including numbers of atoms is shown in S1 Table.

Equilibration of each system was initiated by minimization of the solvated systems using a 500 step SD minimization followed by a 500 step ABNR minimization in which mass-weighted harmonic restraints of 500 kcal/mol/Å$^2$ were applied on the non-hydrogen atoms of the non-loop regions and 100 kcal/mol/Å$^2$ on the non-hydrogen atoms of the loop regions. This was followed by a second round of 500 step SD and 500 step ABNR minimization in which mass-weighted harmonic restraints of 100 kcal/mol/Å$^2$ were applied on the non-hydrogen atoms of non-loop regions and 10 kcal/mol/Å$^2$ on the non-hydrogen atoms of the loop regions. The final minimization involved 500 SD and 500 ABNR steps with 10 kcal/mol/Å$^2$ and 0.5 kcal/mol/Å$^2$ on non-hydrogen of non-loop and loop regions, respectively. Further equilibration and production simulations were performed with OpenMM 7.4[46]. A gradual MD equilibration was necessary to avoid introducing structural distortion in the systems that were still present following the above loop reconstruction and minimization protocols. As most of the protein structures are not involved in the interactions between the Fc and EndoS2, a stronger positional restraint was applied to these regions, while a weaker positional restraint was used and gradually removed on the loop regions described above participating in interactions between the Fc and EndoS2. The protocol for gradual damping of the restraints is shown on S2 Table. Equilibration was performed in the NPT ensemble at 298 K and 1 atm using a Langevin integrator and Monte Carlo barostat for 500 ps with a time step of 1 fs. Further equilibration and production were performed in the NPT ensemble using a time step of 2 fs. LJ interactions were truncated at 11.0 Å with switching initiated at 9.0 Å. Electrostatics were modeled using the particle mesh Ewald method with a real-space cutoff of 11.0 Å. All other simulation parameters were set to default. Snapshots were collected every 5 ps and all simulations were performed with covalent bonds to hydrogens constrained using the SHAKE algorithm.[50]

Hydrogen-deuterium exchange (HDX) protection factors were computed as a function of residue $i$, following the method of Best and Vendruscolo [66]:

$$\ln(\boldsymbol{PF_i}) = \beta_c N_i^c + \beta_h N_i^h \tag{1}$$

In Eq (1) $N_i^c$ is the number of heavy atoms contacts between amide hydrogen of residue i and other residues, $N_i^h$ is the number of hydrogen bonds formed by the amide hydrogen of residue i, and $\beta_c$ and $\beta_h$ are two adjustable parameters with values 0.35 and 2.0, respectively. Specifically, the fraction of deuterium uptake D in the simulation was predicted in a given experimental exposure time t, to depend on the protection factor of that residue, $PF_i$, according to Eq (2): [25, 67]

$$D_i(t) = 1 - exp^{-\left[\left(\frac{k_i^{int}}{PF_i}\right)*t\right]} \tag{2}$$

where $k_i^{int}$ is the intrinsic exchange rate, deduced from experimental measurements for every amino acid type, and depends on the amino acids adjacent in the sequence.[68, 69] For the estimation of intrinsic exchange rate, a temperature of 298 K and pH of 7.4 were used based on both experimental and simulation conditions.[25] The time trace of deuterium fraction of peptide fragment j is taken as an average of the residue-based deuterium fractions: [25, 67]

$$D_j(t) = \frac{1}{n_j}\sum_{i=1}^{n_j}D_i(t) \tag{3}$$

where $n_j$ is the number of amide hydrogens in peptide fragment j.

Contributing to the PF is the number of non-hydrogen atoms $N_i^c$ within a distance of 6.5 Å from the amide nitrogen of residue $i$ calculated using the CHARMM '*coor contact*' command. A cutoff of 2.4 Å between the donor hydrogen and the acceptor was used for identifying number of hydrogen bonds, $N_i^h$, without an angle criterion using CHARMM '*coor hbond*'. Covalently linked neighbor residues were excluded from the calculation for both the number of non-hydrogen atoms and hydrogen bonds.

## Supporting information

Figs include the distribution of distance between N297 and the center-of-mass of CBM binding site and GH active site, definitions of the metrics defining the conformation of EndoS2 including distance, angle and pseudodihedral angle, 2D heatmap plot of the distance between CBM and GH versus a pseudodihedral angle from the apo-EndoS2 simulations, the CBM-modeled complexes, the CBM glycan binding site, cartoon representations of the initial and final frames of selected simulations of each model, models of the full IgG antibody on the final frame from selected simulations of each model, calculated and experimental hydrogen-deuterium exchange protection factors of EndoS2 from the diglycosylated EndoS2-Fc complex, calculated and experimental hydrogen-deuterium exchange protection factors of Fc from the diglycosylated EndoS2-Fc complex and calculated hydrogen-deuterium exchange protection factors Fc from the diglycosylated EndoS2-Fc complex mapped onto Fc. Tables including information: the simulation systems and restraints and the CHARMM loop reconstruction input workflow are presented.

## Supporting information

**S1 Fig. Distribution of distances between N297 and the center-of-mass of A) CBM glycan binding site or B) GH active site.** Red and blue lines correspond to N297 from Fc chain A and B, respectively. Distributions of initial 60,000 complex structures are represented with solid lines and those of the 10,750 structures remaining after the first center-of-mass filter are represented by dotted lines.
(TIF)

**S2 Fig. Annotation of distance, angles and pseudodihedral angle defining the location of the CBM relative to the GH domain (A-D) and (E) a 2-dimensional plot of the minimum distance between the CBM and GH (A) and the pseudodihedral angle (panel D) combined from the 4 apo-EndoS2 simulations (4 simulations for a total of 8 μs).** The (A) distance was based on the minimum distance between the CBM and GH non-hydrogen atoms, angle B was based on the Ca atoms of residues 712 (on CBM), 548 (on LLR) and 186 (on GH), angle C was based on Ca atoms of residues 639 (on hybrid-Ig domain), 548 (on LLR) and 186 (on GH), and the pseudodihedral angle D was defined by the Ca atoms of residue 712 (on CBM), 808 (on CBM), 508 (on LLR) and 106 (on GH). As is evident sampling of shorter distances between the CBM and GH in the closed state is associated with pseudodihedral angles approaching 0˚, while the larger distances associated with the open state correspond to larger pseudodihedral angles in the range of 30 to 50˚. This shows that conformational difference between the open and closed states of EndoS2 have significant contributions from rotation of the CBM around the hinge between the CBM and hybrid-Ig.
(TIF)

**S3 Fig. A. Aggregated initial orientations of the CT glycan surrounding the CBM.** Blue cartoon represents the CBM protein and the glycans are shown as sticks in the initial locations approximately 10 Å from the protein surface used to initiate the SILCS-MC calculations from

which a putative glycan binding site on the CBM was identified as described in the computational methods. **B-D. CBM binding pocket identified based on experimental data and SILCS-MC docking of glycans with the CBM.** B) Top 15 and C) top 6 SILCS-MC docked conformations based on the LGFE scores, and D) amino acids identified based on experimental data [25] (yellow carbons) and those identified based on the 6 lowest LGFE SILCS conformations and hydrogen bond analysis (green carbons).
(TIF)

**S4 Fig. Shift of the Fc on EndoS2 from the selected MD simulations of model A, model B, model C and model D.** White cartoon represents the EndoS2 with the CBM (blue) and the GH (orange) based on the initial simulation 2 coordinates used to start the complex simulation (except model C where simulation 3 is presented). The initial orientation of the Fc in the models used to initiate the MD simulation are shown in green cartoon and the orientation of the Fc from the 2 μs time frame is shown in red cartoon following alignment of EndoS2 to the initial coordinates excluding CBM in the alignment due to its rotation during the MD simulation. Glycans in the initial orientation are shown as purple sticks as in the final orientation as blue sticks. For visualization EndoS2 was removed from the final frame.
(TIF)

**S5 Fig. A. Full Antibody-EndoS2 complex based on model A.** The mAb structure is based on PDB 1IGT [31] following RMSD alignment of the nonhydrogen atoms in the Fc from the model A simulation. The Fc is pink, Fabs are green, GH is orange, CBM is blue and the remainder of EndoS2 is gray. The image indicates that the model A structure can accommodate the full antibody structure given the flexibility of the linker between the Fc and Fabs. **B. Full Antibody-EndoS2 complex based on model B.** The mAb structure is based on PDB 1IGT [31] following RMSD alignment of the nonhydrogen atoms in the Fc from the model B simulation. The Fc is pink, Fabs are green, GH is orange, CBM is blue and the remainder of EndoS2 is gray. The image indicates that the model B structure can accommodate the full antibody structure given the flexibility of the linker between the Fc and Fabs. **C. Full Antibody-EndoS2 complex based on model C.** The mAb structure is based on PDB 1IGT [31] following RMSD alignment of the nonhydrogen atoms in the Fc from the model C simulation. The Fc is pink, Fabs are green, GH is orange, CBM is blue and the remainder of EndoS2 is gray. The image indicates that the model C structure can accommodate the full antibody structure given the flexibility of the linker between the Fc and Fabs. **D. Full Antibody-EndoS2 complex based on model D.** The mAb structure is based on PDB 1IGT [31] following RMSD alignment of the nonhydrogen atoms in the Fc from the model D simulation. The Fc is pink, Fabs are green, GH is orange, CBM is blue and the remainder of EndoS2 is gray. The image indicates that the model D structure can accommodate the full antibody structure given the flexibility of the linker between the Fc and Fabs.
(TIF)

**S6 Fig. Simulation vs experimental hydrogen-deuterium exchange percent differences for A) model A, B) model B, C) model C and D) model D for the diglycosylated species only for all four models.** Differences in the experimental percent deuteration for peptides from EndoS2 in the unliganded and IgG1-complexed states [25] over the first 10s are plotted as blue bars. The analogous differences in the MD calculated percent deuteration for peptides from the EndoS2-apo simulations and from the Fc-glycan-EndoS2 model complex states as red bars. The different domains of Endos2 are labelled across the top of the figure and separated by purple dashed lines as follows: Glycoside hydrolase (GH), leucine-rich repeat (LRR), hybrid-Ig, and carbohydrate-binding module (CBM). Individual peptides are plotted on the

X-axis from the N- to C-terminus based on the sequence number of the first residue in the peptide.
(TIF)

**S7 Fig. Simulation vs experimental hydrogen-deuterium exchange for A) model A, B) model B, C) model C and D) model D for the diglycosylated species only for all four models.** Differences in the experimental percent deuteration for peptides from Fc in the unliganded and IgG1-complexed states [25] over the first 10s are plotted as blue bars. The analogous differences in the MD calculated percent deuteration for peptides from the Fc apo simulations and from the Fc-glycan-EndoS2 model complex states as red bars. Individual peptides are plotted on the X-axis from the N- to C-terminus based on the sequence number of the first residue in the peptide. Note the change in the Y-axis in panel D versus panels A, B and C.
(TIF)

**S8 Fig. Calculated differences in hydrogen–deuterium exchange between apo-Fc and Fc-glycan-EndoS2 complex mapped onto a surface representation of the Fc (PDB ID 1IGT).** [31] The predicted peptide hydrogen-deuterium exchange is average of diglycosylated Fc for A) model A, B) model B, C) model C and D) model D.
(TIF)

**S1 Table. Summary of the simulation systems and simulation times used to generate conformations for EndoS2-Fc-glycan model reconstruction and final EndoS2-Fc-glycan complex.**
(DOCX)

**S2 Table. Harmonic restraints applied during the equilibration MD simulations of the full EndoS2-Fc-glycan model complexes in aqueous solution.** Restraints were only applied to non-hydrogen atoms and units are kcal/mol/Å$^2$.
(DOCX)

**S1 Text. Workflow 1: CHARMM code for loop reconstruction.**
(PDF)

## Acknowledgments

Computational support from the University of Maryland Computer-Aided Drug Design Center is acknowledged.

## Author Contributions

**Conceptualization:** Eric J. Sundberg, Alexander D. MacKerell, Jr.

**Data curation:** Asaminew H. Aytenfisu.

**Formal analysis:** Asaminew H. Aytenfisu, Daniel Deredge, Alexander D. MacKerell, Jr.

**Funding acquisition:** Eric J. Sundberg, Alexander D. MacKerell, Jr.

**Methodology:** Asaminew H. Aytenfisu, Alexander D. MacKerell, Jr.

**Project administration:** Alexander D. MacKerell, Jr.

**Resources:** Alexander D. MacKerell, Jr.

**Supervision:** Alexander D. MacKerell, Jr.

**Visualization:** Asaminew H. Aytenfisu, Daniel Deredge.

**Writing – original draft:** Asaminew H. Aytenfisu, Alexander D. MacKerell, Jr.

**Writing – review & editing:** Asaminew H. Aytenfisu, Daniel Deredge, Erik H. Klontz, Jonathan Du, Eric J. Sundberg, Alexander D. MacKerell, Jr.

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
