## [Decision Letter · Decision Letter 0]

31 May 2021

Dear Prof. MacKerell,

Thank you very much for submitting your manuscript "Insights into substrate recognition and specificity for IgG by Endoglycosidase S2" for consideration at PLOS Computational Biology. As with all papers reviewed by the journal, your manuscript was reviewed by members of the editorial board and by several independent reviewers. The reviewers appreciated the attention to an important topic. Based on the reviews, we are likely to accept this manuscript for publication, providing that you modify the manuscript according to the review recommendations.

Sincerely,

David van der Spoel

Associate Editor

PLOS Computational Biology

Arne Elofsson

Deputy Editor

PLOS Computational Biology

[LINK]

Reviewer's Responses to Questions

**Comments to the Authors:**

Reviewer #1: In this manuscript the authors present the results of an exceptional study of the deglycosylation of IgG Fc-glycans by Endo S2, generating and examining an impressive set of catalytically-competent complexes between an IgG Fc and Endo-S2. In this work, different molecular simulations approaches have been integrated harmoniously and performed successfully, in my opinion, to provide us with much needed insight into the Endo-S2 enzymatic activity. I truly enjoyed reading the manuscript and first and foremost would like to congratulate the authors on such brilliant work. I also would like to bring up the following few points and make some suggestions that the authors may find useful to consider and that I think may help bring the results together into a potential mechanism.

As the authors are aware, in isolated IgGs the two Fc-glycans are tightly packed within the Fc “horseshoe” structure, with each arm (considering complex N-glycans in human IgG1 for example) extending on either side of the Fc (see for example Harbison and Fadda, Glycobiology (2020) doi: https://doi.org/10.1093/glycob/cwz101). The crystal structure of the Endo-S2 in complex with the N-glycan (PDB 6MDS for one) was obtained with isolated N-glycans, i.e. not bound to the Fc. In view of this interactions, I believe, or as a general choice of strategy, molecular docking was used as the first step in making the models, by docking isolated N-glycans and then linking the Fc, if I understood correctly. Because the whole N-glycans do not extend at the sides of the Fc, so are not exposed, yet, as I mentioned earlier, extend across the Fc. Within this framework, I was wondering if the authors noticed in any of their simulations the interaction of only one of the arms on either glycans with the CBM, which in my opinion could potentially initiate extraction. More specifically, if the 1-6 on the CH2-CH3 side facing the domain interacts with the CBM, it could potentially trigger the opening/loosening of the Fc structure, increasing the accessibility to both glycans and promoting the binding of the whole glycan to the CBM and of the other glycan to the GH. This scenario would agree with model D, where the CBM acts as a ‘grip’ facilitating the removal of the opposite N-glycan by GH. The second deglycosylation event could occur according to model C, where the N-glycan bound to the CBM could be ‘transferred’ to the GH, which I found fascinating!

I understand that the above is a mechanistic speculation, yet a plausible one based on the evidence presented in this work and published in the literature, in my opinion, unifying all the different scenarios the authors examined and could be presented in the discussion. In any case, I think it would be useful to comment on how the N-glycans are potentially extracted from within the Fc to bind the CBM and GH.

As minor points,

• I find that it would be really helpful to have Figures presenting the structures of the complexes in the main manuscript, indicating the positions/contacts of the glycans with CBM and GH in different models. Those could be integrated in Figure 1.

• Page 10 and throughout “long-time” MD simulations is probably not a specific term, consider multi-microsecond MD simulations or MD simulations in the low microsecond time range.

• Table 2 caption, “fist glycan” typo

• Page 12, “S2A to D Fig.” probably better as “Fig. S2A to D.”

• Figure 3 caption, the following sentence is unclear to me, please consider revising “Dashed lines indicate....”

• Page 17, “an increase in ~400 Å” units needs to be squared.

Reviewer #2: Let me first make one thing clear, I'm not a computational biologist, but very much interested in immunoglobulin glycosylation and bacterial modification of the functionally important Fc glycans. EndoS2 represents one such very specific strategy with hydrolysis of these glycans, and only when presented in the context of the CH2/CH3 domain of IgG. Some of the authors of the current study have successfully solved the crystal structure of EndoS2 and presented convincing data that both the glycoside hydrolase (GH)domain and the carbohydrate binding domain(CBM) are crucial for the activity on the Fc glycan. Further site directed mutagenesis the solvent exposed site chain of W712 in the CBM results in loss of activity.

If I understand the advanced modeling scheme, known crystal structures of EndoS and IgG Fc are used to investigate the following:

1. Do the CBM and GH interact with IgG i sequence or at the same time?

2. Do the CBM and GH simultaneously interact with the same IgG Fc and/or individually with the two Fc portions within the same IgG molecule?

3. Do the CBM and GH interact with the glycan and/or the protein backbone of CH2/CH3?

4. Can the glycan be transfered from the CBM to the GH and thereby form a catalytically active complex?

The modeling answers these question with that EndoS2 initially interacts with IgG through the CBM followed by interaction with GH to allow for hydrolysis in the chitiobiose core. Furthermore, it is suggested that EndoS2 can adopt both a closed and a more open conformation allowing the CBM and GH to either interact with the same heavy chain or with the two separate heavy chains within the same IgG molecule. Simulations also predict interaction with both the glycan and the protein backbone in the CH2/CH3 domain, and that the Fc glycan can transfer from the CBM within one EndoS2 and thereby facilitate enzymatic activity.

The results from the modeling is compared and consistent with previously presented hydrogen/deuterium exchange data, as well as with previous experimental data indicating the very high specificity of EndoS2 for IgG Fc glycans.

Taken together, the simulations beautifully presents a very plausible model for the detailed interactions between EndoS2 that also fits with earlier experimental findings. However, again I must reveal my somewhat poor understanding of the details of the modeling; is it possible to do some kind of negative control in the modeling (or is it already there?)? For instance, can you do in silico mutations of the solvent exposed side chains in the CBM, or test the know mutations in the GH that leads to loss of activity or a shift towards glycosyl transferase activity?

I have no criticism of the language, introduction to the field, the discussion, or appropriate acknowledgment of previous findings.

**Have the authors made all data and (if applicable) computational code underlying the findings in their manuscript fully available?**

Reviewer #1: Yes

Reviewer #2: Yes

PLOS authors have the option to publish the peer review history of their article (what does this mean?). If published, this will include your full peer review and any attached files.

Reviewer #1: **Yes: **Elisa Fadda

Reviewer #2: **Yes: **Mattias Collin

Figure Files:

Data Requirements:

Reproducibility:

References:

---

## [Editor Report · Decision Letter 1]

30 Jun 2021

Dear Prof. MacKerell,

We are pleased to inform you that your manuscript 'Insights into substrate recognition and specificity for IgG by Endoglycosidase S2' has been provisionally accepted for publication in PLOS Computational Biology.

Best regards,

David van der Spoel

Associate Editor

PLOS Computational Biology

Arne Elofsson

Deputy Editor

PLOS Computational Biology

---

## [Editor Report · Acceptance letter]

13 Jul 2021

PCOMPBIOL-D-21-00876R1 

Insights into substrate recognition and specificity for IgG by Endoglycosidase S2

Dear Dr MacKerell,

I am pleased to inform you that your manuscript has been formally accepted for publication in PLOS Computational Biology. Your manuscript is now with our production department and you will be notified of the publication date in due course.

With kind regards,

Zsofi Zombor
